# Widespread erosion on high plateaus during recent glaciations in Scandinavia

Jane L. Andersen [1], David L. Egholm [1], Mads F. Knudsen[1], Henriette Linge[2], John D. Jansen [1], Vivi K. Pedersen[2], Søren B. Nielsen[1], Dmitry Tikhomirov[3], Jesper Olsen[3], Derek Fabel [4] & Sheng Xu[4]

Glaciers create some of Earth's steepest topography; yet, many areas that were repeatedly overridden by ice sheets in the last few million years include extensive plateaus. The distinct geomorphic contrast between plateaus and the glacial troughs that dissect them has sustained two long-held hypotheses: first, that ice sheets perform insignificant erosion beyond glacial troughs, and, second, that the plateaus represent ancient pre-glacial landforms bearing information of tectonic and geomorphic history prior to Pliocene–Pleistocene global cooling (~3.5 Myr ago). Here we show that the Fennoscandian ice sheets drove widespread erosion across plateaus far beyond glacial troughs. We apply inverse modelling to 118 new cosmogenic $^{10}$Be and $^{26}$Al measurements to quantify ice sheet erosion on the plateaus fringing the Sognefjorden glacial trough in western Norway. Our findings demonstrate substantial modification of the pre-glacial landscape during the Quaternary, and that glacial erosion of plateaus is important when estimating the global sediment flux to the oceans.

[1] Department of Geoscience, Aarhus University, Høegh-Guldbergsgade 2, Aarhus C 8000, Denmark. [2] Department of Earth Science, University of Bergen, and Bjerknes Centre for Climate Research, Bergen N-5020, Norway. [3] Department of Physics and Astronomy, Aarhus University, Ny Munkegade 120, Aarhus C 8000, Denmark. [4] Scottish Universities Environmental Research Center, AMS Laboratory, East Kilbride G75 0QF Scotland, UK. Correspondence and requests for materials should be addressed to J.L.A. (email: jane.lund@geo.au.dk)

Ice has profoundly reshaped pre-glacial landscapes, as shown by the distinct glacial topography of many mid-latitude mountain ranges. However, at higher latitudes, glacial troughs and fjords are in many cases separated by extensive plateaus that, despite being overridden repeatedly by ice sheets, often display few signs of recent glacial erosion[1–7]. Indeed, many plateaus host blockfield mantles indicative of prolonged subaerial weathering[8,9]. In addition, the plateau surfaces frequently contain high abundances of cosmogenic nuclides produced in situ in the upper few metres of Earth's surface by secondary cosmic rays[2–8]. Preservation of cosmogenic nuclides formed during earlier ice-free periods confirms the lack of erosion by recent ice sheets on the plateaus. These observations have been used to suggest that high-latitude plateaus were protected from erosion by cold-based ice sheets over millions of years[2–8]. In contrast, a recent modelling study of ice sheet erosion on passive margins suggests substantial (>100 mm kyr$^{-1}$), but decelerating, erosion rates on plateaus through the late Cenozoic, leading to a gradual emergence of the observed bimodal erosion pattern across the troughs and plateaus[10,11]. Distinguishing between these scenarios is essential, not only to clarify the origin of the plateau landscapes found in many high-latitude regions on Earth, including Scandinavia, Scotland, Greenland, eastern Canada and Antarctica. It is also critical for disentangling the contribution from these extensive regions to the global sediment flux. This aspect is important in light of the recent discourse on the influence of late Cenozoic climatic cooling on the average pace of Earth surface erosion[12–16]. As plateau landforms cover extensive parts of high-latitude continental margins, even small rates of plateau erosion could produce significant sediment contributions.

A key region in which contradicting views on late Cenozoic plateau erosion have collided is western Scandinavia[11,17–22], where summit plateaus have traditionally been interpreted as remnant palaeosurfaces with a pre-Quaternary origin[18–20]. Several studies have applied a mass-balance approach to resolve the problem. Hinging upon the selective linear erosion paradigm[1,23], total glacial erosional output has been estimated by subtracting the present-day topography from the reconstructed palaeosurface envelope projected between the present plateau surfaces[20]. However, revisiting this approach, a recent study found a large mismatch in the sediment volumes reconstructed from fjord erosion relative to the Pliocene–Pleistocene glaciogenic sediments offshore southern Norway[11]. To balance the offshore sediment budget, an additional ~100–400 m of plateau erosion is required[11]. Yet, others reject the need for significant plateau erosion by retrieving the missing sediment volume from the inner shelf and coastal zones[17].

Most plateau surfaces in southern Norway display some signs of glacial erosion (e.g., streamlined bedrock and lake basins), but it remains controversial whether this glacial imprint represents superficial scouring[17–20,24,25] or deeper long-term reshaping by ice sheets[11,21]. Without thorough quantification of recent glacial erosion rates, it is impossible to distinguish between end-member models for the impact of glaciations on these plateau surfaces.

In this study, we set out to resolve recent erosion rates on plateau surfaces by measuring in situ produced cosmogenic nuclide abundances along an ~200 km transect near Sognefjorden in southern Norway (Figs. 1 and 2). To constrain potentially complex exposure and burial histories, we have measured $^{10}$Be and $^{26}$Al abundances in bedrock (60 samples) and boulder erratics (9 samples; Supplementary Table 1–3). We apply a Markov chain Monte Carlo inversion approach[26] to this data set in order to quantify the pattern and depth of erosion on plateau surfaces along the fjord. The modelled erosion rates vary along our transect from >30 m Myr$^{-1}$ close to the coast to ~2–6 m Myr$^{-1}$ on the highest inland plateau sites. We also find that the

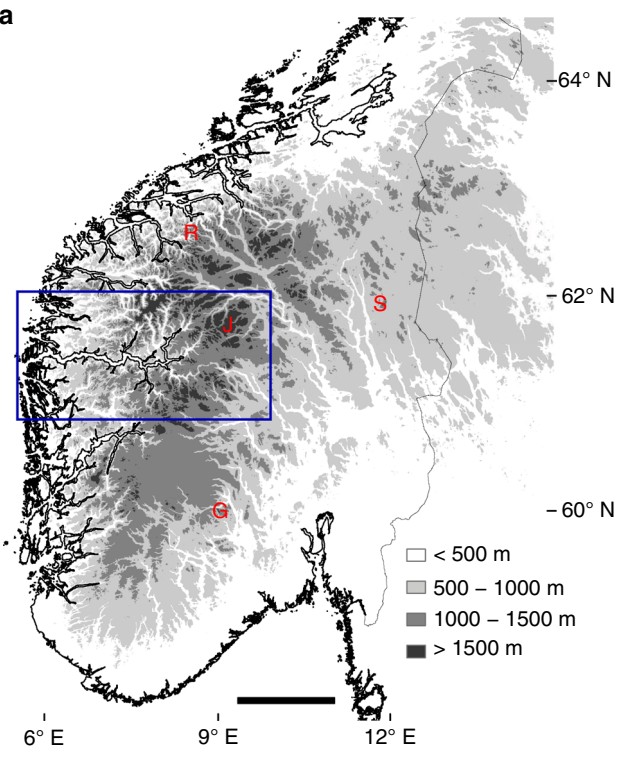

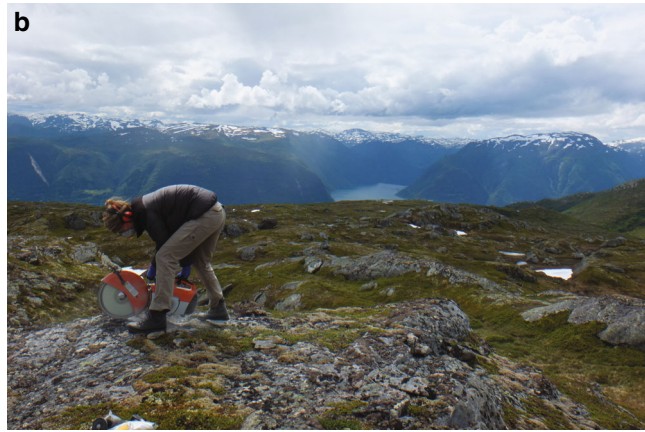

**Fig. 1** Study area and topography in southern Norway. **a** Generalised topography of southern Norway, showing our Sognefjorden study area (blue rectangle, enlarged in Fig. 2) and high isolated peaks: Gaustatoppen (G, at 1883 m a.s.l.) and Sølen (S, at 1755 m a.s.l.). Widespread high-elevation plateaus dissected by deep, narrow fjords characterise the terrain; alpine topography is limited to the highest mountain massifs in Jotunheimen (J) and the Sunnmøre-Romsdal Alps (R). Present-day glaciers cover <1% of the land surface, but during the Last Glacial Maximum the Fennoscandian Ice Sheet buried the entire landscape, possibly excluding a few nunataks[34]. The scale bar is 100 km wide. Map created with ESRI ArcGIS software from a digital elevation model freely available at www.geonorge.no. **b** Sampling an exposed bedrock surface (sample SF44) on a high-elevation plateau with Sognefjorden in the background below

contribution of sediment from the plateaus relative to the total sediment flux is at least 10% when integrating our results over the Quaternary period.

## Results

**A transect of cosmogenic $^{10}$Be and $^{26}$Al along Sognefjorden.** Sognefjorden is the largest fjord system in Scandinavia, reaching

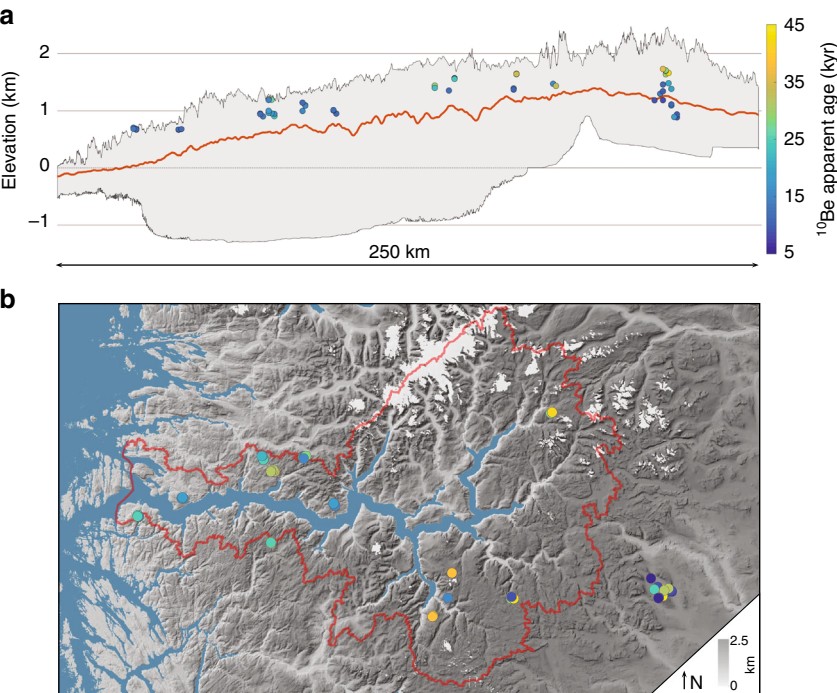

**Fig. 2** Sognefjorden topography and [10]Be apparent exposure ages. **a** West–east topographic swath (140 km wide) along Sognefjorden, showing maximum and minimum elevations (grey band), mean elevations (red line), and our 69 new [10]Be apparent ages (filled circles). **b** Map view of topography around Sognefjorden drainage basin (red line), showing our sample sites (filled circles, legend as above) and present-day glaciers (white). Map created with ESRI ArcGIS software from a digital elevation model freely available at www.geonorge.no

>1300 m below sea level, and its associated ice stream drained much of the southwestern sector of the Fennoscandian Ice Sheet. According to ice-flow models[10,27], topographic steering would drive selective linear erosion and cause ice to thin across the adjacent plateaus. Although most plateaus along Sognefjorden could be classified as areally scoured[28], surface characteristics of the plateaus vary from glacially sculpted sites near the coast to blockfields with few signs of glacial plucking on the highest inland sites. The bedrock samples in our transect span elevations from 660 m above sea level (m a.s.l.) near the coast to 1730 m a.s.l. at the site furthest inland, and have apparent [10]Be exposure ages ranging from 8 to 43 kyr (Supplementary Table 1; Fig. 3a). We primarily targeted bedrock outcrops on top of the plateaus. However, for the easternmost locality, we also collected an elevation profile on bedrock from the plateau summit down to an adjacent glacial trough. The apparent exposure ages within this vertical profile follow a similar elevation trend as samples derived from plateau summits along the transect, for which elevation decreases towards the coast (Fig. 3a). Most boulder erratics and more than one-third of our bedrock samples yield apparent exposure ages that overlap (within 1σ) the timing of the last deglaciation, according to recent compilations (~9–12 kyr)[29,30]. For the glacially sculpted sites closest to the west coast, all apparent [10]Be exposure ages match the deglaciation age (within 1σ), whereas nuclides inherited from previous interglacial periods seem to be present in one or more samples at all other sample sites (Fig. 2).

**Plateau erosion rates derived from inverse modelling.** To examine the longer-term history of erosion beneath glacial ice, we exploited the different decay rates of [10]Be (half-life ~1.39 Myr) and [26]Al (half-life ~0.702 Myr). We measured both [10]Be and [26]Al on 47 bedrock samples and two boulder erratics. To estimate rates of erosion from the [10]Be and [26]Al abundances, we applied a Markov chain Monte Carlo inversion approach[26] to determine

the most likely erosion rates averaged over the past 1 Myr (see Methods). These results provide the first direct and quantitative constraints on long-term erosion rates on southern Norway's high-elevation plateaus. Modelled median rates of erosion vary from 2.4 m Myr$^{-1}$ (interquartile range: 1.2–5.1 m Myr$^{-1}$) to 93 m Myr$^{-1}$ (interquartile range: 34–258 m Myr$^{-1}$; Supplementary Table 3). Overall, the rates of erosion decrease with increasing elevation, although some low-elevation sites also feature slow (<15 m Myr$^{-1}$) erosion (Fig. 4). There is also a systematic west–east gradient in the rate of erosion such that pre-LGM cosmogenic nuclide inventories were more-or-less removed during the last glaciation at sites near the west coast, while some degree of inheritance seems to occur as elevations increase east-ward and inland (Fig. 3a). Note that for samples with apparent [10]Be exposure ages overlapping the deglaciation age[29,30] within uncertainties (see Methods), our inverse model is limited to resolving a minimum estimate of erosion rate (Fig. 4, dashed lines).

Although the plateaus in southern Norway clearly erode more slowly than the adjacent glacial troughs, they are in general eroding relatively fast when compared to a global compilation of cosmogenic nuclide-derived erosion rates from non-glacial settings[31]. Our slowest rates thus approximate the reported global median for bedrock outcrops of 5.4 m Myr$^{-1}$. The modelled rates of erosion show only limited relation to surface characteristics on the sampled plateaus, as overlapping erosion rates were obtained from blockfield-covered (4-6 m Myr$^{-1}$) and glacially sculpted (2–93 m Myr$^{-1}$) sites (Fig. 4, Supplementary Fig. 4). Nevertheless, it is clear that erosion rates obtained from the blockfield-covered site generally are low compared to the wide range of rates obtained for glacially sculpted sites.

The LGM ice sheet was evidently more erosive on lower-lying plateaus towards the western ice margin than on the higher plateau surfaces inland. Nonetheless, our results show that the pre-glacial landscape was modified by substantial glacial erosion,

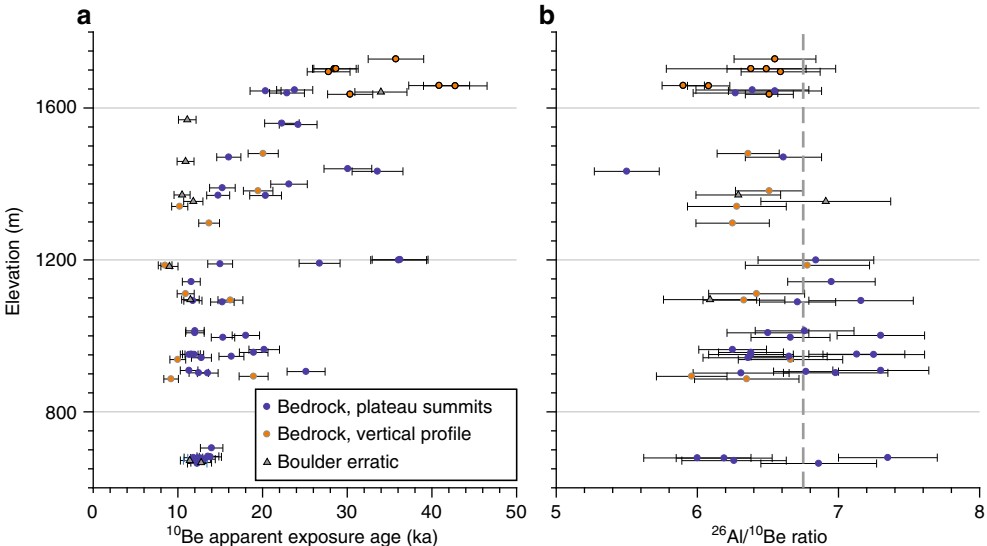

**Fig. 3** [10]Be apparent exposure ages and [26]Al/[10]Be ratios. **a** [10]Be apparent ages ( ± 1σ) from bedrock (n = 60) and boulder erratics (n = 9) relative to elevation. Samples are divided into plateau summits (blue) and a vertical profile from Storlifjell (orange), the easternmost sampled locality. Storlifjell samples >1600 m a.s.l. (circles with black outline) are from the blockfield-covered plateau summit. **b** Forty-nine [26]Al/[10]Be ratios ( ± 1σ), relative to the surface production ratio, 6.75 (dashed grey line)

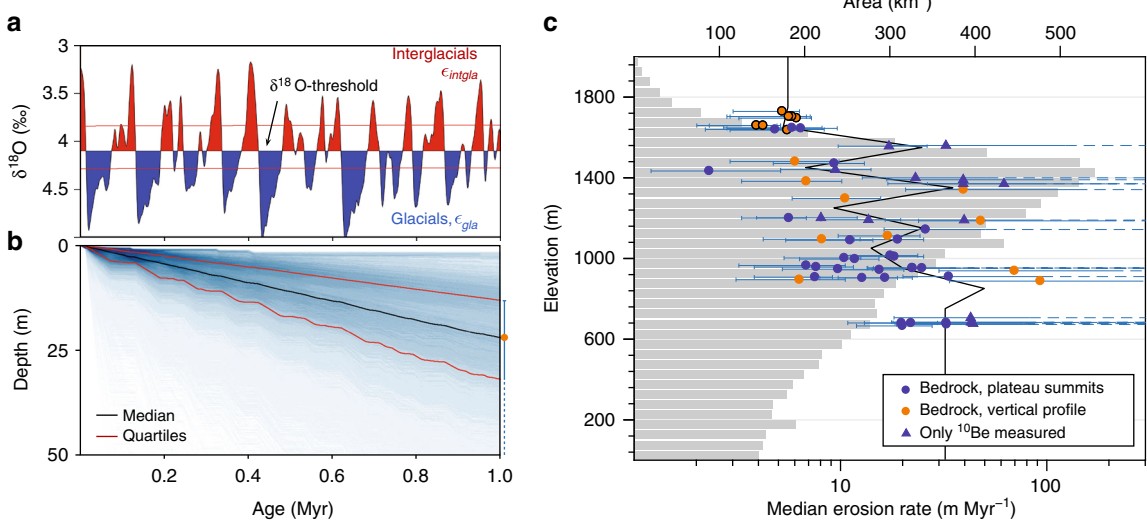

**Fig. 4** Inverse-modelled erosion rates. **a** Conceptual representation of the Markov chain Monte Carlo (MCMC) model, defining glacial and interglacial periods from a stacked benthic δ[18]O record[49], which is a proxy for global ice volume. The δ[18]O-threshold is a free model parameter, and red lines indicate a typical uncertainty associated with this parameter. **b** Example of a modelled exhumation history (sample SF53) since 1 Myr. The blue shades represent the distribution of 200,000 simulated exhumation histories consistent with measured nuclide abundances. Modelled median erosion rate (orange circle) and uncertainty (blue bar) for this sample is indicated on the right. **c** Modelled median erosion rates (m Myr[−1], quartile uncertainties, bottom axis) for all available [10]Be and [26]Al pairs (circles) and [10]Be measurements (triangles). Vertical profile samples >1600 m a.s.l. (circles with black outline) are from a blockfield-covered plateau summit. Maximum erosion rates are unconstrained for samples without inheritance (dashes to the right). Grey bars represent the hypsometry of the Sognefjorden drainage basin (top axis). Black wiggle links the mean erosion rates binned at 100 m interval

a process that may have started prior to the onset of the Quaternary. The widespread erosion of plateau surfaces along Sognefjorden, which exceeds 10 m Myr[−1] in most places (62 % of samples) and 30 m Myr[−1] in many cases (27 % of samples; Fig. 4c), thus seems incompatible with the notion of pristine paleosurfaces left untouched by glacial erosion[20]. On the other hand, our erosion rates are generally lower than the proposed average plateau erosion rate of 130 m Myr[−1] over the past 2.8 Myr derived from an onshore–offshore mass balance study[11].

## Discussion

The apparent cosmogenic [10]Be and [26]Al exposure ages measured in bedrock samples from plateaus surrounding Sognefjorden are younger than those reported from mountain peaks elsewhere in southern Norway[32–34] (Supplementary Fig. 1). Cosmogenic nuclide abundances show that these, often blockfield-covered peaks were not deeply eroded during the most recent glacial cycles[32–34]. Yet, very high apparent [10]Be exposure ages in southern Norway seem to be confined to summits, like

Gaustastoppen (>70 kyr) and Sølen (>160 kyr)[33], which stand high above the plateaus. In contrast, bedrock samples from upland plateaus in eastern Canada[2,4,5,8], Greenland[7], and northern Scandinavia[3,6] often yield high apparent cosmogenic nuclide ages and low $^{26}$Al/$^{10}$Be ratios. This indicates limited plateau erosion by recent ice sheets in these regions. Thus, cold-based conditions on summits and high plateaus are often associated with scenarios where ice sheets mainly erode in troughs, just as proposed by the selective linear erosion concept[1,17,23]. The results presented here do not oppose the concept of selective linear erosion, but demonstrate that high plateaus are not necessarily associated with non-erosive ice. In fact, our results contribute towards quantifying and understanding the degree of selectivity associated with glacial erosion of passive margins. Our results also open the possibility that high plateaus in Arctic regions, which currently experience low erosion rates, eroded more efficiently under former ice sheets prior to the periods that can be constrained by cosmogenic nuclides[10].

We propose that vast high-elevation, low-relief areas in southern Norway were not covered by cold-based and non-erosive ice sheets throughout the Quaternary. Instead, our results imply that these areas could have contributed significantly to the net delivery of sediments to the oceans and the proposed global trend of increased late Cenozoic sedimentation[12,14,35–37]. To provide a first-order assessment of the contribution from these plateau surfaces, we estimate the total Quaternary erosion output from the part of the Sognefjorden drainage basin that is above sea level (see Methods). Interpolating our modelled erosion rates, we find a total sediment yield of 650 km$^3$. Previous estimates of fjord excavation in Sognefjorden are ~6000–8000 km$^3$ (refs. [11,20]); hence, we suggest that plateau erosion accounts for ~10 % of the total sediment flux. We note that this estimate is less than that previously suggested for plateau erosion in western Scandinavia (~45–65 %) based on the mass balance between fjord excavation and offshore sediment volumes from the late Pliocene and Quaternary (last 2.8 Myr)[11]. However, at least three factors may work towards converging these estimates. Firstly, for one-third of our samples, our method is limited to resolving erosion rate minima; hence, our sediment yield estimate is a minimum. Secondly, our estimated fractional contribution from plateaus would likely be larger if integrated over all of southern Norway, since the fjord-to-plateau surface area ratio is higher in the Sognefjorden drainage basin than in the wider region. Finally, inherent to our inversion modelling is an approach known as 'two-stage uni-formitarianism'[26] in which previous glacial periods share a common erosion rate (i.e., varying from place to place but constant through time) and likewise for interglacial periods. The assumption that recent glaciations were equally effective at eroding the plateaus is justifiable over a restricted timescale, although we anticipate that glacial trough-deepening over the Quaternary may have steered progressively more ice discharge into the fjords and away from plateaus[10,27]. Given that thin ice promotes less erosive, frozen-bed conditions, this development likely caused a gradual deceleration of glacial erosion over the plateaus[10]. Considering these aspects, our modelled predictions of the pace of plateau erosion should be regarded as minimum estimates. We expect that as glacial troughs deepened over time, the disparity in erosion rates between troughs and plateaus became more pronounced. Resolving whether these factors can bridge the gap between disparate plateau erosion estimates requires more widespread quantification of erosion rates in western Scandinavia, preferably on longer timescales. However, the results presented in this study clearly demonstrate that the plateaus in western Scandinavia are actively eroding landscapes and as such are continuously remoulded by surface processes. Inferences about pre-glacial topography based on the present-day

morphology should thus be filtered for the effect of glacial (and periglacial) erosion during the Pleistocene.

## Methods

**Sampling.** We targeted high-elevation low-relief areas with quartz-bearing lithologies spanning a large area and wide elevation range along Sognefjorden. Bedrock samples were collected in sets of two to six per site, each sample spaced by a few metres to some hundred metres apart to evaluate local variations in cosmogenic nuclide abundances. Wherever possible, one to two glacial boulder erratics were sampled at the same sites. The lithologies in the area are predominantly granitic to dioritic gneisses. Depending on the estimated quartz content, 1–4 kg of rock was collected, typically by cutting out surface blocks with a diamond-blade rocksaw, but in some instances by manually chiselling quartz veins. Site location, shielding measurements, surface inclination, sample thickness and surface attributes (lithology, lichen cover, geometry and degree of weathering) were recorded in the field. We preferentially aimed for local high points in the landscape to minimise cosmic-ray shielding due to snow and sediment cover. The majority of samples had minimal topographic shielding, only a thin lichen cover and were taken from flat and horizontal rock surfaces with comparatively wide fracture spacing.

**Laboratory Procedures.** All rock samples were crushed and sieved to extract the 250–500 μm size fraction. To isolate quartz, a series of mineral separation processes were applied, including aqua regia leaching, floatation, magnetic separation and boiling in phosphoric acid[38]. Quartz was then sequentially leached in 2% HF/2% HNO$_3$ for a minimum of $3 \times 3$ days. Be and Al isotope extraction chemistry was conducted at the Scottish Universities Environmental Research Centre (SUERC), Glasgow, and at the university laboratories in Aarhus and Bergen, following standard procedures[38,39]. One processing blank followed each batch of 9–15 unknowns. About 200–260 μg of Be carrier (Scharlau BE03450100 at SUERC; Scharlau BE03460100 at Aarhus) was added to each sample and the processing blank. On the basis of ICP-OES (inductively coupled plasma optical emission spectrometer) analysis of the quartz, the approximate native Al content was determined and, if necessary, samples were spiked with Al carrier (Fischer Scientific ICP solution at SUERC; SPEX ICP solution at Aarhus) to reach a total Al content of ~1000–1500 μg. Following digestion in concentrated HF, we removed an aliquot of the samples for ICP-OES analysis, typically ~4–6 % of the total volume. From this aliquot, the total Al and Be content of the sample was determined. Samples were then dried down and converted to chloride form, and Be and Al were isolated and purified via ion chromatography. After oxidation, BeO was mixed with Nb-powder (ratio: 1:4–1:6) and Al$_2$O$_3$ was mixed with Ag-powder (ratio: 1:1–1:2) and then pressed into copper cathodes ready for AMS (accelerator mass spectrometry).

**Cosmogenic Nuclide Concentrations.** These are presented in Fig. 2, Supplementary Tables 1–3 and Supplementary Fig. 2. Isotopic ratios were analysed at the Aarhus AMS centre (AARAMS; $n = 41$) or at SUERC ($n = 28$; supplementary Table 1). At AARAMS, $^{10}$Be/$^9$Be and $^{26}$Al/$^{27}$Al isotopic ratios were measured with a multielement AMS system[40]. The $^{10}$Be/$^9$Be ratios of measured samples vary from $0.110 \times 10^{-12}$ to $1.190 \times 10^{-12}$, with errors ranging from 1.5 to 3.9% (1 SD, $n = 41$). Be results were normalised to ICN standard 01-5-4 with a $^{10}$Be/$^9$Be ratio of $2.851 \times 10^{-12}$ (ref. [41]; sample errors include 1.1% uncertainty of the ICN standard). The average machine background was $5.45 \pm 2.95 \times 10^{-16}$ (1 SD, $n = 5$), while the average processing blank ratio was $2.4 \pm 1.7 \times 10^{-15}$ (1 SD, $n = 7$). Energy spectra at the rare isotope detector indicated efficient separation of $^{10}$Be from $^{10}$B; therefore, no isobaric corrections are required. The $^{26}$Al/$^{27}$Al ratios of measured samples vary from $0.407 \times 10^{-12}$ to $4.860 \times 10^{-12}$, with errors ranging from 1.8 to 5.0% (1 SD). Al results were normalised to ICN standard 01-4-3 with a $^{26}$Al/$^{27}$Al ratio of $10.65 \times 10^{-12}$ (ref. [42]; sample errors include 1.2% uncertainty of the ICN standard). The average machine background was $9.17 \pm 4.81 \times 10^{-16}$ (1 SD, $n = 6$), while the average processing blank ratio was $3.3 \pm 1.8 \times 10^{-15}$ (1 SD, $n = 7$). At SUERC, $^{10}$Be/$^9$Be and $^{26}$Al/$^{27}$Al isotopic ratios were measured with a 5 MV NEC accelerator mass spectrometer[43]. The $^{10}$Be/$^9$Be ratios of measured samples varied from $7.8 \times 10^{-14}$ to $1.1 \times 10^{-12}$, with errors ranging from 1.5 to 5.0% (1 SD, $n = 28$). Be results were normalised to NIST SRM4325, with a ratio of $2.79 \times 10^{-11}$ (ref. [41]). The average processing blank ratio was $7.2 \pm 6.1 \times 10^{-15}$ (1 SD, $n = 2$). The $^{26}$Al/$^{27}$Al ratios of measured samples vary from $1.7 \times 10^{-12}$ to $4.7 \times 10^{-12}$, with errors ranging from 2.4 to 2.9% (1 SD, $n = 8$). Al results were normalised to Z92-0222, with a ratio of $4.11 \times 10^{-11}$. The average processing blank ratio was $9.9 \pm 7.1 \times 10^{-15}$ (1 SD, $n = 2$).

For both $^{10}$Be and $^{26}$Al samples, we subtracted the processing blank ratio from sample ratios on a batch-by-batch basis and propagated uncertainties including balance error. The processing blank data used to perform blank correction for each sample is reported in Supplementary Table 2. $^{26}$Al blank corrections were insignificant for all samples (0.1–1.1 %), while $^{10}$Be blank corrections were insignificant for most samples (0.1–2.3%), whereas a few were more significant, although not enough to affect our data interpretations: SF01 (3.8%), SF26 (14.7%) and SF35-SF41 (4.3–8.0%).

For $^{26}$Al measurements, the total Al content (native + carrier), determined by ICP-OES, was used to calculate the $^{26}$Al concentration. For samples prepared in

Bergen and Aarhus, we assessed the accuracy of the total Al determination by performing repeat measurements of an independent multielement ICP-standard. Results typically fell within a few percent of the certified value. The actual vs. expected content in the processing blank samples yielded an average of 99.5% ($n = 7$), while the concurrent measurement of Be in processing blanks and samples yielded an average of 97.5% ($n = 48$) of the expected content. For the Al samples prepared at SUERC ($n = 8$), we unfortunately do not have independent quality control of the ICP measurements. However, the ratios obtained from these samples do not stand out compared to samples prepared in Aarhus and Bergen, and removing them from the analysis would have no bearing on our conclusions.

**Calculation of Apparent exposure ages**. All apparent exposure ages derive from the online calculators formerly known as the CRONUS-Earth online calculators' source code version 2.3 (ref. [44]; http://hess.ess.washington.edu) constants file 2.3. We used a global calibration of the $^{10}$Be spallation production rate[45] and the Lal/ Stone time-independent scaling scheme[46,47] (reference spallation production rate 4.01 atoms g$^{-1}$ yr$^{-1}$). Our apparent exposure ages are not corrected for glacioisostatic uplift or snow cover, which might lead to overestimation of the production rates and thus a slight underestimation among our reported ages. The effect of glacioisostatic adjustments is negligible compared to uncertainties associated with other factors, such as temporal changes in air pressure[48]. Moreover, almost nothing is known about isostasy during prior glacial cycles; hence, manipulating production rates based solely on Holocene postglacial rebound is likely to misrepresent samples with significant nuclide inheritance. Present-day snow cover is highly variable between our sample sites (www.senorge.no). The effect of snow shielding is less significant for sites close to the coast, but >80 km inland snow thicknesses >1 m are likely to persist for >5–6 months per year. Assuming 1 m half-time snow cover with a relatively high density of 0.3 g cm$^{-3}$, $^{10}$Be production rates decrease by ~8 %, while a 2-m snow cover for 8 months of the year would result in a production rate decrease of ~20%. Owing to limited knowledge of the long-term changes in snow distribution for both this study area and for production rate calibration sites, we ignore these effects but minimise them by selecting samples on topographic high points.

**Markov chain Monte Carlo Inverse Model**. Details of our inversion approach are documented in Knudsen et al.[26]. The method entails repeated forward simulations of $^{10}$Be and $^{26}$Al production on bedrock surfaces during ice-free periods coupled with removal of these nuclides due to erosion and radioactive decay. Glacial (total burial under ice, i.e., zero production of $^{10}$Be and $^{26}$Al) and interglacial (ice-free) periods are defined via a threshold applied to a stacked benthic $\delta^{18}$O record[49], which is a proxy for changes in global ice volume and temperature during the past few million years. The $\delta^{18}$O-threshold, the interglacial and glacial erosion rates, and the timing of the last deglaciation, are represented by free parameters in the model, and each sample is modelled by four random walkers until a total of 200,000 ($4 \times 50,000$) simulations have been accepted[26]. The estimated erosion rate is based on the median exhumation history of all accepted forward simulations integrated over the past 1 Myr; uncertainties derive from the first and third quartiles. All model results are presented in Supplementary Table 3. Although the site-specific deglaciation age is a free model parameter, it is allowed to vary only within ±1000 years of the deglaciation age reported in a recent review of the Fennoscandian deglaciation[30]. This choice is supported by the observation that eight out of nine boulder erratics and more than one-third of all bedrock samples in our data set overlap with the regional deglaciation ages within 1$\sigma$[29,30]. Relaxing this constraint in the model to 0–20 kyr for deglaciation yields a slight increase (1–2 m Myr$^{-1}$) in median erosion rates among bedrock samples with the slowest erosion rates and produces unconfined maximum erosion rates for a larger fraction of the samples (Supplementary Fig. 3). The value of the $^{26}$Al/$^{10}$Be surface production ratio is widely assumed to be 6.75 (e.g., ref. [44]). Yet, as this value is currently debated[50], we document the effect of using an alternative ratio of 7.3 in Supplementary Fig. 3. For most of our samples, a higher surface production ratio leads to a slight decrease (1–2 m/Myr) in median erosion rate estimates.

**Quaternary sediment yield from Sognefjorden**. We calculated the average erosion rate within 100 m elevation bins for our data set and multiplied those values by the terrain surface area within each elevation interval in the Sognefjorden drainage basin over the Quaternary (2.6 Ma). We assumed steady erosion rates, corresponding to the boundary values, for elevations outside our data set range (7.9 m Myr$^{-1}$ for elevations >1750 m a.s.l., 84 m Myr$^{-1}$ for elevations 0 – 650 m a.s.l.).

**Data availability**. The authors declare that the main data supporting the findings of this study are available within the paper and its Supplementary Information. Extra data and model codes are available from the corresponding author on request.

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

## Acknowledgements

We acknowledge funding from the Danish Council for Independent Research (grant DFF-6108-00226) and Aarhus University Research Foundation. J.D.J. was supported by the Australian Research Council (DP130104023) at the University of Wollongong and by a Marie Skłodowska-Curie Fellowship at the University of Potsdam. V.K.P. acknowledges financial support from the Research Council of Norway. We thank Maria M. Rodriquez for careful guidance in preparation of the first batch of cosmogenic nuclide samples and Lars Evje for conscientiously performing the ICP analyses at University of Bergen. We thank the reviewers for highly constructive and helpful reviews.

## Author contributions

J.L.A., D.L.E., S.B.N. and M.F.K. designed the study. J.L.A., D.L.E., M.F.K., V.K.P. and J.D.J. collected the samples. J.L.A. and H.L. prepared the samples. D.T. and J.O. performed AMS measurements at AARAMS, S.X. and D.F. at SUERC. J.L.A. and D.L.E. carried out the data analysis. All authors contributed to the writing of the paper.

## Additional information

**Competing interests:** The authors declare no competing interests.

