## [Peer review file · Nature Communications]

Reviewers' comments:

Reviewer #1 (Remarks to the Author):

This is an important paper in that it brings together a wealth of new cosmogenic data to quantify the role of glaciers in shaping a plateau landscape in Norway. The wider importance comes from the widespread existence of plateau landscapes inboard of passive continental margins. What emerges from the abundant and clearly presented cosmogenic data is that there has been significant glacial erosion of the plateau remnants and that it increases from ~ 5m on the highest uplands to ~40 m at the coast. They can even place limits on the erosion achieved by the last glacial cycle. The paper effectively links the cosmogenic data to wider models of landscape evolution during Cenozoic glaciations involving both offshore sediment volumes and other geomorphological data. The figures are excellent and clear. Altogether I think they have established an important benchmark for the way Cenozoic glaciations in the northern hemisphere have eroded passive margins.

My criticism would be that they could put this across more directly. At present the results are framed as a challenge to a local (Norwegian?) understanding of the hypothesis of selective glacial erosion. Rejection of the hypothesis leads to the view that the flat nature of plateau remnants are not relict fluvial landscapes but somehow linked to glacial and periglacial action. My interpretation of their results is different. It is that they have contributed significantly to our understanding of how ice sheets in a maritime setting modify ancient erosion surfaces. Such a study helps enormously in interpreting the subglacial landscapes (and even ice sheet stability) in Antarctica and Greenland.

The early work on landscapes created by glaciers envisaged a continuum between a plateau with no erosion, through light areal scouring to heavy areal scouring and this is seen for example in early maps and analyses of landscapes modified by the former Laurentide ice sheet (eg. J. Glac. 1978, 20 (83), 367-391). Selective linear erosion describes a category of landscape in highlands where sometimes there are no landforms of glacial erosion on the plateau even when adjacent to a deep trough. Such plateau remnants often bear tors, smooth fluvial slopes and even a regolith of Tertiary age. Subsequent cosmogenic work in Sweden, Baffin Island and Antarctica seems to support such an interpretation, but only for surfaces with no landforms of glacial erosion.

The photos show that the plateaus in Norway sampled in this study comprise a landscape of areal scouring with ice-moulded landforms clearly visible. Even the photo of the blockfield seems to show the outline of eroded bedrock. Thus, the authors are not working on a pristine landscape of selective linear erosion, but on a glacially modified one. And this is their great achievement – to quantify how much has been removed and how the pattern increases towards the coast! Moreover, the discovery of rates of glacial erosion of only <5m/Ma next to a fjord extending 2 km below the plateau is some geomorphological feature and, indeed, a demonstration of selectivity!

The final thrust of the paper is to suggest that the evidence of erosion in the last few million years under glacial and periglacial conditions is enough to create the flat plateau surfaces. The authors recognise that they are contradicting other Scandinavian work. Is it wise to set up such a challenge unless backed up by theory or firm evidence? The first issue is that there is no known process by which an ice sheet could create flat surfaces; in contrast, we know that fluvial action in continental areas can create flat surfaces and that such surfaces exist in the non-glacial world.

Secondly, here in Norway there is an ancient fluvial signal visible in the dendritic pattern of Sogne Fjord; thus it would seem likely that the flatness of the plateau surfaces could likewise be an inheritance from the fluvial past. The authors could address this latter comment by simply omitting the last paragraph and the last line of the abstract.

Detail.

Title. Good

Abstract. The challenge to a selective linear erosion paradigm seems a diversion from their great results. Why not stress that for the first time you quantify the depth and pattern of glacial erosion of a passive continental margin?

Line 40. This interpretation of the established hypothesis needs more clarification. Comparing a pristine example in Sweden with a glacially sculpted one in Norway is comparing two quite

different situations.

Lines 51-63. I liked this context.

Lines 86-87. Could you explain the range in altitudes more fully? I assume all are on plateau surfaces, and that the lower sites are towards the coast. Somewhere it would help to describe the plateau sites in a more detail. For example, I notice that there is considerable consistency within sites with depths of erosion calculations showing a contrast between $\sim 5\text{m/Ma}$ in the east (Storifjell), 9-30 m in the middle at Aurland, and $>40\text{ m}$ near the coast at Gulen. What is it about these sites that explains the consistency and what causes the greater variability elsewhere? And later I was puzzled by the nature of the vertical transect. What plateau sites are at low altitude at the inland site of Storifjell?

Line 128. What is meant by a varying degree of inheritance? Is it less?

Lines 152-155. Yes, here you agree with the selective hypothesis in these areas.

Lines 158-160. Most would agree.

Lines 165-169 and 171-203. I liked the widening horizons here.

Lines 205-220. I am unconvinced by your arguments here. See above.

Line 213. The word peneplain has implications linked to the early 20th Century work of Davis.

Many geomorphologists would prefer to use the more objective term: erosion surface.

Fig 3. Caption. I was puzzled by the vertical profile from Storifjell. Are some samples taken from fjord sides and valleys? Are they all on plateau remnants?

Fig 4 Caption. Puzzled by the blue shading which is densest along the line of the upper quartile and also at the upper limit of the distribution. Why choose SF53?

Reviewer #2 (Remarks to the Author):

Review of:

Widespread erosion on high plateaus during recent glaciations in Scandinavia

by Andersen et al.

Nature Communications, October 2017

In this manuscript, the authors use numerical models and new cosmogenic nuclide data to study glacial landscape evolution. Specifically, they assess plateaus and challenge long-held paradigms about how such landforms evolve subglacially. The authors conclude that plateaus in Scandinavia do not represent relict surfaces as traditionally thought; rather, they have been glacially carved and likely date to the Quaternary.

Overall, I would like to commend the authors on a job well done. The study is useful for understanding how landforms evolve subglacially, and does a good job of putting the work in context of what has been done previously. The authors effectively set up hypotheses and demonstrate how their work challenges the generally-accepted paradigm of selective linear erosion. It is also worth mentioning that this is an impressively large new dataset of multi-isotope cosmogenic nuclide measurements and will add significantly to the literature.

However, I think the paper could benefit from revision. The message of the paper would benefit from clarification and streamlining, and there are areas that are in need of more detail. Certain assumptions and subtleties should be made transparent for the reader, since as a cosmogenic community I think we want to be upfront about the limitations of the technique.

Below, I have detailed several major suggestions as well as a number of minor suggestions. Again, I commend the authors on an interesting study and am happy to see so many new Nordic cosmogenic datapoints.

Lee Corbett

Ashley.Corbett@uvm.edu

Big-picture comments:

Overall significance: At present, I find the paper would benefit from more explicit links to the big picture. These plateaus represent a relatively small area. Why should readers of Nature Communications care about them? How might understanding these plateaus help us think about glaciation or erosion in a much larger sense? Drawing these links to larger-scale questions and processes is critical to making your work relevant to an interdisciplinary audience. The manuscript eventually gets there, but not until the last paragraph; and the big picture linkages in that last paragraph are primarily focused on landscape development whereas much of the introduction is focused on sediment flux, so the message isn't coherent. The paper would be greatly improved by having a more streamlined and consistent message that is set up at the beginning and carried throughout.

Inheritance presence/absence: Determining whether a surface contains inherited cosmogenic nuclides is not black and white. There are uncertainties associated with the exact timing of glacial retreat, as well as analytic uncertainties associated with the cosmo measurements themselves. The reality is that it's impossible to make a clear distinction between which samples contain inherited nuclides and which do not. This needs much more finesse in the manuscript, particularly lines 91-98. Right now, I worry it will be misleading to readers.

Deciding which samples have been buried: I'm not convinced your use of two-sigma (see lines 104-106) is appropriate. I understand why you provide both one-sigma and two-sigma numbers for the reader, but because $^{26}\text{Al}/^{10}\text{Be}$ uncertainties are larger than for a single isotope, the reality is that two sigma is quite far out from continuous exposure. There may be many samples between one and two sigma that preserve real burial, maybe even up to a few hundred ka worth (although a sensitivity test would assess this more quantitatively). In short, you may be rejecting real burial. Another way of looking at this would be to see how many samples are one-sigma and two-sigma above constant exposure, since presumably the ratios shouldn't be allowed to be that high and hence might be giving you a more realistic picture of statistical spread. Similar to the above, I think this piece is drawing conclusions that will be misleading to the reader. There is no magical cut-off here, and I feel strongly this should be treated more cautiously and with more transparency.

Transferability: The ideas you propose here are interesting, and they provide clarity about your landscape of interest. But the paper does not explicitly address how transferable the approach and findings are to other landscapes (which may help with the "big picture", see above). Is what you learned specific to Norway? Could it also apply to Greenland? What about cold-based ice landscapes in Antarctica? What factors might dictate why or why not your conclusions are transferable? You get there to some extent on Lines 185-188, but I would like to hear more.

Blanks: Your blanks are confusing since it sounds like you prepared samples in three separate labs and analyzed them on two different AMS systems. This needs to be made much more transparent for the reader. In the Methods, please detail exactly how many process blanks you have from which sets of conditions, and how you corrected. For example, were samples prepped at SUERC and analyzed at SUERC corrected using only the blanks from the same set of conditions? It's not clear as written. Further, the backgrounds you report seem to only be for the Aarhus AMS, not for SUERC, and you refer to them as "machine blanks". This term usually implies a direct precip blank that has not been processed along with your samples. I assume you also ran process blanks and used the process blanks to correct the samples? This section needs to be clearer and the blank corrections may need some rethinking. I acknowledge the blank correction is quite small proportionally for these samples, but it still should be made clear to the reader.

Minor comments:

Line 21: Add how many measurements you made; it's an impressive number but is buried about 80 lines into the paper.

Line 41: I'm not sure "assumed" is the right word here since many of these studies are based in field observations and data. Maybe "inferred"?

Line 43: Should be "suggests" instead of "suggest" since it's a singular study.

Line 43: I don't think "significant" is the right word here. It's vague, and it also implies some sort of statistical meaning. I'd prefer to see an actual rate (or range of rates?).

Lines 45-49: I agree with you that your work is important for understanding sediment fluxes. But the sediment flux piece seems to come out of nowhere, since the abstract and the whole first paragraph are focused on landscape development rather than sediment generation. See the major comment about finding a coherent big picture.

Lines 61-63: Again, I agree with you that this is important for thinking about sediment fluxes. But make the link back to the big picture for the reader here. Why should we care about sediment flux? Why is it important?

Line 73: "Recent" is vague here- clarify.

Line 76: I don't think you need the hyphen after exposure.

Line 90: You refer the reader to the supplement for existing age control on the termination of the last glaciation, but at least tell the reader here where that age control is coming from. Is it radiocarbon?

Line 115: Here and throughout, your inferred erosion rates need to have uncertainties. At present, you give the reader no sense for how well-constrained these are and what assumptions have gone into the calculations.

Line 118: Clarify how you define "overlapping"- indistinguishable at one-sigma?

Line 124: Is it really fair to compare your erosion rates to all non-glacial settings? The Portenga paper looks at erosion rates all around the world, from a huge variety of climates. I'm not saying this comparison can't be made, but it really needs to be clarified for the reader. I seem to remember that Portenga sub-divided based on climate classifications. How do your rates compare with other cold landscapes? And is the value you quote on line 125 just bedrock, or both bedrock and basins?

Line 128: Should be "occurs".

Lines 137-140: See earlier comment about needing uncertainties on all of these rates.

Line 140: Clarify what "average" this is. For where? Is this averaged over the whole landscape, i.e. both plateaus and troughs?

Line 160-162: I'm struggling with how to interpret this sentence. The "elsewhere" is vague, and I'm having trouble with the 1 Myr time period. This is less than two half-lives for ^{26}Al , so I'm not clear on why you're drawing a cut-off so recently. Do you mean that all the erosion on the plateaus is old?

Line 169: "Recent" here is vague.

Line 180: It's unclear what the 45-65% is. Is this the portion of plateau sediment contribution in relation to the whole?

Lines 216-220: I like these ideas- they make an intriguing, broad hook for your paper. See my major comment about the big picture. As a reader, it was frustrating to wait until the last few sentences of the paper to see the larger significance, and seemed like an inconsistent message because much of the introduction is focused on sediment yields. The paper would flow more effectively if there were a more focused, coherent message throughout.

Line 286: When you refer to minimal shielding, do you mean topographic?

Lines 306-307: I would like to see a few more sentences about your ^{27}Al quantification. How did you assess the accuracy of your ICP work? Did you use an internal standard? How well do the expected and actual ^{27}Al contents of your blanks compare? Your ^{26}Al data are only ever as good as your ^{27}Al quantification, so these are important details to include for the reader.

Figure 2: I assume the colored dots in the second panel are the ^{10}Be ages, color-coded the same way is in the first panel? Consider rewording the caption and potentially moving the legend so that it's clear what those dots are.

Figure 4: For panel A, clarify in the caption that the 180 threshold is something you've chosen and imposed upon the model. For panel B, what are the blue bar and orange circle on the right side? For panel C, you might specify which axes the different parameters are plotted against since it's a complex figure.

Reviewer #3 (Remarks to the Author):

This paper presents results from paired ^{10}Be and ^{26}Al cosmogenic isotope analyses for bedrock surfaces (60) and boulders (9) from glacially-eroded terrain that stands above the magnificent glacial trough of the Sognefjord and its tributaries. The main conclusions of the study are that (1) high-elevation plateaus can form as a result of glacial and periglacial processes within the Quaternary, and (2) need not be relict landforms inherited from a pre-Quaternary landscape and (3) do not represent dissected remnants of once conjoined, preglacial surfaces that have been left largely unmodified by ice sheets throughout the Quaternary.

The authors present important new data on the depths and rates of glacial erosion for a swath of terrain standing above the Sognefjord and rising W-E. This is a product of a bold sample campaign which yielded an unusually large cosmogenic isotope sample set. The paper is clearly written and the diagrams are generally very good. I suggest however that the new results do not adequately support the conclusions.

The concept of selective linear glacial erosion is now almost 50 years old 1. David Sugden recognised that in the Cairngorms Mountains, Scotland, ice sheets had covered the terrain and selectively carved deep troughs out of a pre-existing surface which, in immediately adjacent areas, it has left barely modified. On the Cairngorm plateau surfaces, landforms of glacial erosion such as roches moutonnées and rock basins are absent from large areas and non-glacial landforms, notably tors, are common, rising out slopes that also carry blockfields and, locally, saprolites. In contrast, in the Cairngorm glens glacial erosion has been highly effective, with deepening of precursor fluvial valleys by a maximum of 350 m 2. Glacial erosion here has been selective and linear because it was at least an order of magnitude greater in depth along the valleys.

Sugden elegantly linked form to process. He reasoned that to excavate deep glacial valleys the ice moving within the valleys must be erosive whereas the preservation of fragile tors on plateaux required that here ice was non-erosive through multiple glacial cycles. He identified the pressure melting point of basal ice as a fundamental threshold for glacial processes. Ice above the PMP could slide and erode whereas ice below the PMP remained frozen to its bed and could not slide and erode, with movement confined to internal deformation and creep.

The Cairngorms became the type area for selective linear glacial erosion because of its striking morphological contrasts. Areas with similar geomorphology were soon identified from the passive margins around the North Atlantic 3 and from the Canadian Arctic 4. Landscapes of selective linear glacial erosion were recognised also as representing a distinctive zone of glacial erosion on passive margins, intermediate in position and erosion depths between the high elevation, intensely dissected rift shoulders of western Scotland and Norway, the west and east coasts of Greenland and the east coast of Labrador and the low elevation, low erosion zones of NE Scotland 5, Finnish Lapland 6, northern Greenland 7 and Baffin Island 8. Low erosion rates indicated by cosmogenic nuclides 7,9 allow that in these landscapes of selective linear glacial erosion plateaux can be largely relict landforms inherited from a pre-Quaternary landscape and so may represent dissected remnants of once conjoined, preglacial surfaces that have been left largely unmodified by ice sheets throughout the Quaternary 10.

This reasoning is broadly matched by arguments presented on Lines 30-39 and 145-155 in the manuscript. On Line 40 however there is a jump to a wider, rather different concept of selective linear glacial erosion. Here we start to deal with landscapes where glacial erosion is widespread – with glacial erosion forms widely developed outside of deep valleys. In the case of the Sognefjord, we are in a zone of high glacial erosion on the shoulder of the Norwegian passive margin, with extensive glacial scour outside of the glacial valleys. Glacial erosion is still very uneven in its impact in such landscapes 11 12, with at least one order of magnitude difference between erosion depths in the valley and those on ridges and hills 13. Selective erosion occurs because glacier ice flows towards valleys and hollows and is thicker in those topographic lows and hence glacial erosion is concentrated in these lows. This selectivity drives the glacial buzzsaw to cut down preferentially in mountainous terrains 14. Mathematical modelling shows how topography acting on glacial dynamics gives contrasts not only in basal ice temperatures but also in sliding velocities and meltwater fluxes within and outside valleys 15.

The Sognefjord is the largest fjord system in Norway and penetrates about 200 km inland from the coast 16. The mountains along Sognefjord rise eastward from about 500 m in the coastal region to reach elevations above 2000 m in Jotunheimen. The glacial valley now occupied by the fjord has been regarded as incised within an older valley form with an original relief of at least several tens of metres 16,17. Across SW Norway and within the Sognefjord basin, stepped erosion surfaces have been identified at elevations of 800-1200, 1300-1400, 1500-1800 and 2000 m a.s.l., each with internal relief of at least many tens of metres 18. During the Pleistocene, ice caps and ice sheets built up repeatedly across the Sognefjord drainage basin, filling the Sognefjord and advancing onto the shelf. During the last glaciation, ice advance occurred at ~34 ka and the Sognefjord basin was probably ice covered for all of the period until ~11 ka, apart from a brief period of marine conditions at the outer coast in the Allerød. The behaviour of ice in Sognefjord was probably highly dynamic in space and time due to the extreme water depth in the fjord and the very high snow accumulation rates on the surrounding high ground 19.

The paper sets up two linked hypotheses to test using erosion rates derived from cosmogenic nuclides: (1) that glacial erosion is minimal beyond glacial troughs and that (2) plateaux represent little modified preglacial landsurfaces. Does the experimental design of the study fully address these hypotheses?

The study area is the high elevation basement terrain that lies within a swath, 5-20 km wide, along the axis of the Sognefjord and its tributaries. Is this a representative case study area? No if comparisons are to be drawn with the Cairngorm type area or with similar terrains with few glacial

landforms on surrounding plateaux. No also if the preservation of little-modified preglacial surfaces is to be examined because the best preserved remnants of such landscapes in Fennoscandia have been mapped previously in zones of lower glacial erosion outside the Sognefjord basin 20,21. But yes if the focus is on valley-upland erosion contrasts on a deeply dissected rift shoulder. And yes because minimal erosion has been claimed for the terrain above the Sognefjord valley 22. Due to the high relief and precipitation in SW Norway compared to the rest of Fennoscandia, it is likely that the volumes of ice discharged via Sognefjord were amongst the highest for outlet glaciers and ice streams along the Norwegian margin. Hence the rates of glacial erosion on the benches and hills alongside the glacial trough can be expected to be amongst the highest for terrain in comparable locations.

The samples are grouped in 10 localities alongside the Sognefjord and its tributaries. Grouping is helpful, as the authors point out, because it provides a window on the variability of cosmogenic nuclide inventories within each area. Little justification however is given in the text for choosing these sample areas. This is odd given the hypotheses to be tested. Where are the sample sites in relation to the supposed valley floor, plateaux and palaeosurfaces? Nesje and others have indicated where they think the former Sogne valley floor was.

Others have mapped palaeosurfaces and relief types across SW Norway, including the Sogne basin 23 20. To test the two hypotheses above it might be expected that sampling would have focussed on these specific locations regarded previously as preglacial topographic remnants. Moreover to test if remnants of precursor topography remain sample sites might include those where the lowest glacial erosion could be predicted. For example, apparently well-preserved plateaux fragments mapped previously. Or sites remote from the main valley axis or in lee locations for ice flow. This seems to me to be a blind-spot in the sampling strategy of the project.

Sample locations are on ridge and hill summits; not all locations can be described as belonging to plateaux, despite repeated, rather vague references to plateaux in the text. Local relief varies from tens to hundreds of metres within the sample areas at scales of a few km². Most locations have been scoured by ice - western locations tend to be glacially streamlined whereas eastern locations are more or less roughened. However morphological differences between areas and sites seen at different scales and elevations in air and ground photos are not assessed in relation to erosion depths in the paper - these differences may be important 24. Some sites retain blockfields, potentially a useful indicator of low intensity glacial erosion if rock weathering in frost-susceptible rock types is not entirely post-glacial, but I don't see where in the paper or supplementary information the blockfield samples are identified or the blockfields described?

What then of the results? Supplementary Figure 2 provides a clear summary of spatial distribution of the ¹⁰Be results. Cosmogenic isotope inheritance is present at all sites except perhaps for two western sites, Lavik and Balestrand. There is some doubt about whether or not slight nuclide inheritance is present because of the wide 9-12 ka range for deglaciation ages. Cosmogenic isotope inheritance requires that <2.5-3 m of rock was removed by the last ice sheet, probably with <1 m at many sites. It would be good to see these calculations for the last glaciation to compare with the longer term modelled erosion rates. For the westernmost site, Gulen, the presence of nuclide inheritance (shown by differences in bedrock and boulder ¹⁰Be ages) is remarkable. This site is on a ridge transverse to ice flow and fully exposed to ice leaving the outer fjord. Yet <3 m of rock was removed from the ridge after ~20 ka of km-thick, warm-based ice flow. Indeed the whole data set is remarkable because it demonstrates widespread cosmogenic isotope inheritance across the Sogne basin, a major outlet for the last FIS.

The Markov-Chain Monte Carlo (MCMC) model approach developed by Knudsen et al. 25 is innovative and important - it allows extrapolation of cosmogenic isotope results back in time. MCMC is based on the assumption that the exposure/burial history can be divided into two distinct regimes: (i) glacial intervals with subglacial erosion and, due to shielding by the overlying ice sheet, no exposure, and (ii) interglacial intervals with subaerial erosion and full exposure, assuming no significant shielding by for example, snow, till, or vegetation. The rates of glacial and

interglacial erosion may vary spatially, but for any particular bedrock sample the two erosion rates are assumed to be uniform throughout all glacials and interglacials, respectively. The MCMC model does not include abrupt individual erosion events, such as subglacial plucking, but integrates the effects of such events over time. In this study, the complication of snow cover is acknowledged in the text and addressed in the supplementary information. A likely error of 8% based on modern snowfall data (Line 356) is suggested, representing >1 m snow cover for 6 months of the year. This may be too conservative for high elevation sites and for periods of cooler climates than present. Greater snow shielding gives more significant over-estimation of erosion rates. Also MCMC assumes that the last glacial phase provides a close analogue for previous phases. This may be true but we know that FIS has switched between ice cap and ice sheet modes through the Pleistocene 26 and we suspect that ice in the Sognefjord drainage basin may behave very differently when the fjord is filled with ice or water. A further complication is that glacially-polished rock surfaces probably are subject to increasing weathering and erosion rates after exposure. It should also be noted that the apparent ^{10}Be ages all fall within the last 50 ka so that an extrapolation to time spans of 1 Ma suggested by expressing erosion rates in units of m/Myr is a long stretch.

The modelled erosion estimates are all for summits and modelled as 2-93 m/Myr. The highest estimates are for samples with low TCN inheritance where the last ice sheet removed >2 m of rock. 73% of samples appear to indicate erosion rates of <30 m/1 Myr. Even at low elevation sites there are samples with rates <15 m/Myr. Erosion rates of <10 m/Myr are recorded from all sites at >800 m elevation. This latter point is crucial because when viewed across extensive rock surfaces and at the Myr timescale it is the slowest erosion rates that matter. If nearby rock surfaces erode at, say, 10 m/Myr and 50 m/Myr then this would lead to increasing relief and leave the first surface upstanding. This topographic relief is not seen on sample summit areas. Also if erosion rates of 50 m/Myr were extensive then all summit sites with slow erosion rates of <10 m/Myr would be destroyed within 1 Myr. The differences in erosion rates within individual area likely relate mainly to what happened to different rock surfaces during the last glaciation – for example whether or not an episode of bedrock quarrying occurred.

Rates of erosion of <10 m/Myr are compatible with the modification of pre-existing valley floors and erosion surfaces with an original relief of many tens of metres. We should remember however that the erosion rates reported here come from summits in hard, relatively unfractured rock. Deepening of surrounding shallow, fracture-guided basins and valleys has involved greater depths and rates of glacial erosion. Hence glacial erosion in successive glaciation can act to lower, modify and destroy older topography. Glaciers tend to destroy rather than create low-relief rock surfaces through the innate tendency for ice flow to be faster, thicker and warmer along valleys 14. Several mentions are made in the text of the formation of plateaux by glacial erosion but relief along the swath sampled in this paper is increasing – the basins and tributary valleys set into the terrain above the Sognefjord are becoming deeper.

The authors are correct to point to significant glacial modification of rock surfaces in the uplands surrounding the Sognefjord. The glacial morphology developed across much of the Sognefjord basin, except its highest ground, makes this manifest. Similar modification has been mapped widely across SW Norway 20. In the lowland basement terrain of southern Sweden, where extensive palaeosurfaces are also mapped 27, Pleistocene glacial erosion of basement is estimated generally as 10-20 m below summits, reaching >50 m in valleys 28. We may be seeing the start of a convergence of FIS glacial erosion long-term average rates in hard bedrock at a magnitude of ~10 m/Myr. The results presented in this paper thus may represent an important contribution to the rejection of the extreme and rather dated end-member views of no Pleistocene modification of palaeosurfaces versus complete glacial eradication of palaeosurfaces presented at the start of this paper. The way is clear for re-adoption of an older, more moderate view 29 30 that widespread topographic inheritance is possible if significant but spatially variable glacial and periglacial erosion is accepted.

The question of sediment volumes offshore remains. The volume of rock evacuated from the Sognefjord was first generated by Nesje and others from a smoothed summit surface using summit heights and then subtracting the present relief from it. This surface subsumes small valleys and rock basins and so this erosion is included in estimates. To this should be added the summit erosion depths estimated by the authors. Perhaps also an allowance for former saprolite layers³¹. That will still leave a large deficit between source and sink – pointing to deep erosion of the coast and the near-shore zone³².

Adrian Hall

References

- 1 Sugden, D. E. The selectivity of glacial erosion in the Cairngorm Mountains, Scotland. *Transactions of the Institute of British Geographers* 45, 79-92, doi:10.2307/621394 (1968).
- 2 Hall, A. M. & Gillespie, M. Fracture control on valley persistence: the Cairngorm Granite pluton, Scotland. *International Journal of Earth Sciences*, doi:10.1007/s00531-016-1423-z (2016).
- 3 Sugden, D. E. Landscapes of glacial erosion in Greenland and their relationship to ice, topographic and bedrock conditions. *Institute of British Geographers Special Publication 7*, 177-195 (1974).
- 4 Sugden, D. E. Glacial erosion by the Laurentide ice sheet. *Journal of Glaciology* 20, 367-391 (1978).
- 5 Hall, A. M. & Sugden, D. E. Limited modification of mid-latitude landscapes by ice sheets: the case of north-east Scotland. *Earth Surface Processes and Landforms* 12, 531-542, doi:10.1002/esp.3290120510 (1987).
- 6 Hall, A. M., Sarala, P. & Ebert, K. Late Cenozoic deep weathering patterns on the Fennoscandian shield in northern Finland: a window on ice sheet bed conditions at the onset of Northern Hemisphere glaciation. *Geomorphology* 246, 472-488 (2015).
- 7 Corbett, L. B., Bierman, P. R., Graly, J. A., Neumann, T. A. & Rood, D. H. Constraining landscape history and glacial erosivity using paired cosmogenic nuclides in Upernavik, northwest Greenland. *Geological Society of America Bulletin* 125, 1539-1553 (2013).
- 8 Ebert, K. GIS-analyses of ice-sheet erosional impacts on the exposed shield of Baffin Island, eastern Canadian Arctic. *Canadian Journal of Earth Science*, doi:10.1139/cjes-2015-0063 (2015).
- 9 Corbett, L. B., Bierman, P. R. & Davis, P. T. Glacial history and landscape evolution of southern Cumberland Peninsula, Baffin Island, Canada, constrained by cosmogenic ¹⁰Be and ²⁶Al. *Geological Society of America Bulletin* 128, 1173-1192, doi:10.1130/b31402.1 (2016).
- 10 Schermer, E. R., Redfield, T. F., Indrevær, K. & Bergh, S. G. Geomorphology and topography of relict surfaces: the influence of inherited crustal structure in the northern Scandinavian Mountains. *Journal of the Geological Society*, doi:10.1144/jgs2016-034 (2016).
- 11 Anderson, R. S., Molnar, P. & Kessler, M. A. Features of glacial valley profiles simply explained. *Journal of Geophysical Research* 111, 1-14 (2006).
- 12 Kessler, M. A., Anderson, R. S. & Briner, J. P. Fjord insertion into continental margins driven by topographic steering of ice. *Nature Geoscience* 1, 365-369 (2008).
- 13 Strunk, A. et al. One million years of glaciation and denudation history in west Greenland. *Nature Communications* 8, 14199, doi:10.1038/ncomms14199 <http://www.nature.com/articles/ncomms14199#supplementary-information> (2017).
- 14 Hall, A. M. & Kleman, J. Glacial and periglacial buzzsaws: fitting mechanisms to metaphors. *Quaternary Research* 81, 189-192, doi:10.1016/j.yqres.2013.10.007 (2014).
- 15 Hall, A. M. & Glasser, N. F. Reconstructing former glacial basal thermal regimes in a landscape of selective linear erosion: Glen Avon, Cairngorm Mountains, Scotland. *Boreas* 32, 191-207, doi:10.1080/03009480310001100 (2003).
- 16 Nesje, A. & Sulebak, J. R. Quantification of late Cenozoic erosion and denudation in the Sognefjord drainage basin, western Norway. *Norsk Geografisk Tidsskrift* 48, 85 - 92 (1994).
- 17 Aarseth, I., Nesje, A. & Fredin, O. West Norwegian Fjords. 45 (*Norsk Geologisk Forening*, 2014).
- 18 Lidmar-Bergström, K., Bonow, J. M. & Japsen, P. Stratigraphic Landscape Analysis and geomorphological paradigms: Scandinavia as an example of Phanerozoic uplift and subsidence.

Global and Planetary Change 100, 153-171 (2012).

19 Mangerud, J., Gyllencreutz, R., Lohne, Ö. & Svendsen, J. I. in Quaternary Glaciations -Extent and Chronology: Part IV. a closer look. (eds J. Ehlers & P Gibbard) 279-298 (Elsevier, 2011).

20 Etzelmüller, B., Romstad, B. & Fjellanger, J. Automatic regional classification of topography in Norway. Norsk Geologisk Tidsskrift 87, 167-180 (2007).

21 Fjellanger, J. & Etzelmüller, B. Stepped palaeosurfaces in southern Norway - interpretation of DEM-derived topographic profiles. Norsk Geografisk Tidsskrift 57, 102-110 (2003).

22 Nesje, A., Dahl, S. O., Valen, V. & Ovstedal, J. Quaternary erosion in the Sognefjord drainage basin, western Norway. Geomorphology 5, 511-520, doi:10.1016/0169-555X(92)90022-G (1992).

23 Green, P. F., Lidmar-Bergström, K., Japsen, P., Bonow, J. & Chalmers, J. A. Stratigraphic landscape analysis, thermochronology and the episodic development of elevated, passive continental margins. Geological Survey of Denmark and Greenland Bulletin 30, 1-150 (2013).

24 Krabbendam, M. & Bradwell, T. Quaternary evolution of glaciated gneiss terrains: pre-glacial weathering vs. glacial erosion. Quaternary Science Reviews 95, 20-42 (2014).

25 Knudsen, M. F. et al. A multi-nuclide approach to constrain landscape evolution and past erosion rates in previously glaciated terrains. Quaternary Geochronology 30, 100-113, doi:10.1016/j.quageo.2015.08.004 (2015).

26 Kleman, J., Stroeven, A. P. & Lundqvist, J. Patterns of Quaternary ice sheet erosion and deposition in Fennoscandia and a theoretical framework for explanation. Geomorphology 97, 73-90, doi:10.1016/j.geomorph.2007.02.049 (2008).

27 Lidmar-Bergström, K., Olovmo, M. & Bonow, J. M. The South Swedish Dome: a key structure for identification of peneplains and conclusions on Phanerozoic tectonics of an ancient shield. GFF, 1-16, doi:10.1080/11035897.2017.1364293 (2017).

28 Lidmar-Bergström, K. A long-term perspective on glacial erosion. Earth Surface Processes and Landforms 22, 297-306, doi:10.1130/G20343.1 (1997).

29 Linton, D. L. The forms of glacial erosion. Transactions of the Institute of British Geographers 33, 1-28, doi:10.2307/620998 (1963).

30 Rudberg, S. Gross morphology of Fennoscandia - six complementary ways of explanation. Geografiska Annaler 70A, 135-167, doi:10.2307/521068 (1988).

31 Glasser, N. F. & Hall, A. M. Calculating Quaternary erosion rates in North East Scotland. Geomorphology 20, 29-48, doi:10.1016/S0169-555X(97)00005-6 (1997).

32 Hall, A. M., Ebert, K., Kleman, J., Nesje, A. & Ottesen, D. Selective glacial erosion on the Norwegian passive margin. Geology 41, 1203-1206, doi:10.1130/G34806.1 (2013).

In this document, reviews are in black with line numbers referring to the originally submitted manuscript, while responses from the authors are in blue with line numbers referring to the revised paper.

Reviewer #1 (Remarks to the Author):

C1-1. This is an important paper in that it brings together a wealth of new cosmogenic data to quantify the role of glaciers in shaping a plateau landscape in Norway. The wider importance comes from the widespread existence of plateau landscapes inboard of passive continental margins. What emerges from the abundant and clearly presented cosmogenic data is that there has been significant glacial erosion of the plateau remnants and that it increases from ~ 5m on the highest uplands to ~40 m at the coast. They can even place limits on the erosion achieved by the last glacial cycle. The paper effectively links the cosmogenic data to wider models of landscape evolution during Cenozoic glaciations involving both offshore sediment volumes and other geomorphological data. The figures are excellent and clear. Altogether I think they have established an important benchmark for the way Cenozoic glaciations in the northern hemisphere have eroded passive margins.

My criticism would be that they could put this across more directly. At present the results are framed as a challenge to a local (Norwegian?) understanding of the hypothesis of selective glacial erosion. Rejection of the hypothesis leads to the view that the flat nature of plateau remnants are not relict fluvial landscapes but somehow linked to glacial and periglacial action. My interpretation of their results is different. It is that they have contributed significantly to our understanding of how ice sheets in a maritime setting modify ancient erosion surfaces. Such a study helps enormously in interpreting the subglacial landscapes (and even ice sheet stability) in Antarctica and Greenland.

Response: First of all, thanks for the very encouraging comments. We have modified the paper to provide a more balanced discussion of selective linear erosion and extended the focus on subglacial landscape formation on passive margins. See also our response to comment C1-2, C1-3 and C1-4.

C1-2. The early work on landscapes created by glaciers envisaged a continuum between a plateau with no erosion, through light areal scouring to heavy areal scouring and this is seen for example in early maps and analyses of landscapes modified by the former Laurentide ice sheet (eg. J. Glac. 1978, 20 (83), 367-391). Selective linear erosion describes a category of landscape in highlands where sometimes there are no landforms of glacial erosion on the plateau even when adjacent to a deep trough. Such plateau remnants often bear tors, smooth fluvial slopes and even a regolith of Tertiary age. Subsequent cosmogenic work in Sweden, Baffin Island and Antarctica seems to support such an interpretation, but only for surfaces with no landforms of glacial erosion.

The photos show that the plateaus in Norway sampled in this study comprise a landscape of areal scouring with ice-moulded landforms clearly visible. Even the photo of the blockfield seems to show the outline of eroded bedrock. Thus, the authors are not working on a pristine landscape of selective linear erosion, but on a glacially modified one. And this is their great achievement – to quantify how much has been removed and how the pattern increases towards the coast!

Moreover, the discovery of rates of glacial erosion of only <5m/Ma next to a fjord extending 2 km below the plateau is some geomorphological feature and, indeed, a demonstration of selectivity!

Response: In response to this comment along with comments of the other reviewers, we have modified the abstract, introduction and discussion part of the paper so as to tighten our definition of selective linear erosion relative to some wider variants that have emerged over time. We acknowledge that there is a large contrast in erosion rates between plateaus and fjords also in the Sognefjord area and that this is in line with the selective linear erosion hypothesis. However, we report plateau erosion rates much faster than those found on other passive continental margins. Our results suggest substantial modification of the plateaus by glacial processes beyond removal of surficial deposits such as blockfields. This has, to our knowledge, not been demonstrated earlier.

C1-3.

The final thrust of the paper is to suggest that the evidence of erosion in the last few million years under glacial and periglacial conditions is enough to create the flat plateau surfaces. The authors recognise that they are contradicting other Scandinavian work. Is it wise to set up such a challenge unless backed up by theory or firm evidence? The first issue is that there is no known process by which an ice sheet could create flat surfaces; in contrast, we know that fluvial action in continental areas can create flat surfaces and that such surfaces exist in the non-glacial world. Secondly, here in Norway there is an ancient fluvial signal visible in the dendritic pattern of Sogne Fjord; thus it would seem likely that the flatness of the plateau surfaces could likewise be an inheritance from the fluvial past. The authors could address this latter comment by simply omitting the last paragraph and the last line of the abstract.

Response: We would like to point out, that although the plateau landscapes around Sognefjord are indeed low-relief, they are by no means flat. Much of this so-called palaeic relief comprises hundreds of metres of relief. Furthermore, recent modelling efforts have shown that both glacial (Egholm et al. 2017) and periglacial (Anderson, *Geomorphology* 46 (2002) p. 35-58, Egholm et al. 2015) processes can lead to the formation of high-elevation low-relief landscapes. However, we understand the reviewers concern that this suggestion is not conclusively backed by data in this study and we have modified the last paragraph accordingly.

Detail.

Title. Good

C1-4. Abstract. The challenge to a selective linear erosion paradigm seems a diversion from their great results. Why not stress that for the first time you quantify the depth and pattern of glacial erosion of a passive continental margin?

Response: We have followed this comment and highlighted this aspect in the abstract. See also response to comment C1-2.

C1-5. Line 40. This interpretation of the established hypothesis needs more clarification. Comparing a pristine example in Sweden with a glacially sculpted one in Norway is comparing two quite different situations.

Response: We are not referring to an example from Sweden in this sentence; it is unclear what the reviewer is referring to here. However, since the other reviewers also commented on this sentence (C2-9 and C3-1) and since we did not feel it added substantially to the message, we have deleted it.

Lines 51-63. I liked this context.

C1-6. Lines 86-87. Could you explain the range in altitudes more fully? I assume all are on plateau surfaces, and that the lower sites are towards the coast. Somewhere it would help to describe the plateau sites in a more detail. For example, I notice that there is considerable consistency within sites with depths of erosion calculations showing a contrast between $\sim 5\text{m/Ma}$ in the east (Storifjell), 9-30 m in the middle at Aurland, and $>40\text{ m}$ near the coast at Gulen. What is it about these sites that explains the consistency and what causes the greater variability elsewhere? And later I was puzzled by the nature of the vertical transect. What plateau sites are at low altitude at the inland site of Storifjell?

Response: We have clarified the explanation of our sampling strategy further, as we understand how this can lead to some confusion. Our primary target in this study was bedrock on broad, convex plateau summits, however, for our easternmost locality (Storlifjell) we also collected an elevation profile on bedrock from the summit down into an adjacent glacial trough. The variability in erosion rates we find at most of our sites is intriguing, especially as the variation do not seem to correspond to surface morphology or variables such as joint spacing within each site. We have strived to make both of these points clearer to the reader and thank the reviewer for this suggestion.

C1-7. Line 128. What is meant by a varying degree of inheritance? Is it less?

Response: The intention was to say that nuclides inherited from prior to the last glacial period seem to be present in one or more samples at all sites further inland. As we can see why this sentence can lead to confusion, we have modified it slightly – see also response to comment C1-6 and C2-4/C2-5.

C1-8. Lines 152-155. Yes, here you agree with the selective hypothesis in these areas.

Response: We do, and we also agree with the hypothesis for the Sognefjord area so far as it suggests higher erosion in the fjord than on the plateaus. However, our results show that erosion on plateaus adjacent to a big fjord can also be substantial, and to our knowledge this has not been demonstrated previously.

Lines 158-160. Most would agree.

Lines 165-169 and 171-203. I liked the widening horizons here.

C1-9. Lines 205-220. I am unconvinced by your arguments here. See above.

Response: See responses to comments C1-1 to C1-4 as well as C2-3.

C1-10. Line 213. The word peneplain has implications linked to the early 20th Century work of Davis. Many geomorphologists would prefer to use the more objective term: erosion surface.

Response: The word 'peneplain' has been extensively used in the Scandinavian context, thus also in some of the references cited here. However, we have removed this sentence as a response to some of the other comments and the issue is no longer relevant here.

C1-11. Fig 3. Caption. I was puzzled by the vertical profile from Storifjell. Are some samples taken from fjord sides and valleys? Are they all on plateau remnants?

Response: We have clarified this in the main text, see also our response to C1-6.

C1-12. Fig 4 Caption. Puzzled by the blue shading which is densest along the line of the upper quartile and also at the upper limit of the distribution. Why choose SF53?

Response: The purpose of this figure (4b) is to illustrate the concept of our modelling approach. This example (SF53) was chosen because it has a modelled erosion rate roughly in the middle of the rates obtained within this study. As the reviewer notes, it also illustrates that the median is not necessarily corresponding to the densest part of the distribution.

Reviewer #2 (Remarks to the Author):

Review of:

Widespread erosion on high plateaus during recent glaciations in Scandinavia
by Andersen et al.

Nature Communications, October 2017

In this manuscript, the authors use numerical models and new cosmogenic nuclide data to study glacial landscape evolution. Specifically, they assess plateaus and challenge long-held paradigms about how such landforms evolve subglacially. The authors conclude that plateaus in Scandinavia do not represent relict surfaces as traditionally thought; rather, they have been glacially carved and likely date to the Quaternary.

C2-1. Overall, I would like to commend the authors on a job well done. The study is useful for understanding how landforms evolve subglacially, and does a good job of putting the work in context of what has been done previously. The authors effectively set up hypotheses and demonstrate how their work challenges the generally-accepted paradigm of selective linear erosion. It is also worth mentioning that this is an impressively large new dataset of multi-isotope cosmogenic nuclide measurements and will add significantly to the literature.

Response: Thanks for the encouraging comments!

C2-2. However, I think the paper could benefit from revision. The message of the paper would benefit from clarification and streamlining, and there are areas that are in need of more detail. Certain assumptions and subtleties should be made transparent for the reader, since as a cosmogenic community I think we want to be upfront about the limitations of the technique.

Response: We have edited the paper according to the suggestions made by this and the other two reviewers and hope that the resulting manuscript conveys our message more clearly. See responses to specific comments below.

Below, I have detailed several major suggestions as well as a number of minor suggestions. Again, I commend the authors on an interesting study and am happy to see so many new Nordic cosmogenic datapoints.

Lee Corbett

Ashley.Corbett@uvm.edu

Big-picture comments:

C2-3. Overall significance: At present, I find the paper would benefit from more explicit links to the big picture. These plateaus represent a relatively small area. Why should readers of Nature Communications care about them? How might understanding these plateaus help us think about glaciation or erosion in a much larger sense? Drawing these links to larger-scale questions and processes is critical to making your work relevant to an interdisciplinary audience. The manuscript

eventually gets there, but not until the last paragraph; and the big picture linkages in that last paragraph are primarily focused on landscape development whereas much of the introduction is focused on sediment flux, so the message isn't coherent. The paper would be greatly improved by having a more streamlined and consistent message that is set up at the beginning and carried throughout.

Response: We have altered the introduction and discussion of the paper to focus on the central result, namely the quantification of the depth and pattern of glacial erosion along a passive continental margin. Furthermore, we have tried to highlight the relation between erosion and sediment flux (which we argue is basically the same thing). We also argue that in previously glaciated arctic areas, plateaus like those addressed here do not represent relatively small areas. On the contrary, they represent in many cases the norm, whereas glacial troughs occupy relatively little area. Hence, constraining erosion on previously glaciated plateaus is very important for our general understanding of glacial landscape evolution at high latitudes. Hopefully these changes succeed in streamlining the message of the paper.

C2-4. Inheritance presence/absence: Determining whether a surface contains inherited cosmogenic nuclides is not black and white. There are uncertainties associated with the exact timing of glacial retreat, as well as analytic uncertainties associated with the cosmo measurements themselves. The reality is that it's impossible to make a clear distinction between which samples contain inherited nuclides and which do not. This needs much more finesse in the manuscript, particularly lines 91-98. Right now, I worry it will be misleading to readers.

Response: We agree with the reviewer on this point. A discussion of the presence or absence of inherited nuclides is not key to our argument and, consequently, we have removed the paragraph that contained a detailed discussion of inheritance. Instead we focus the discussion on our modelled erosion rates, which are more robust outcomes of our study. As our MCMC model systematically addresses the possible scenarios of erosive and non-erosive burial, it deals with inheritance in a more objective manner that incorporates the uncertainties on both CN measurements and assumptions on deglaciation time.

C2-5. Deciding which samples have been buried: I'm not convinced your use of two-sigma (see lines 104-106) is appropriate. I understand why you provide both one-sigma and two-sigma numbers for the reader, but because $^{26}\text{Al}/^{10}\text{Be}$ uncertainties are larger than for a single isotope, the reality is that two sigma is quite far out from continuous exposure. There may be many samples between one and two sigma that preserve real burial, maybe even up to a few hundred ka worth (although a sensitivity test would assess this more quantitatively). In short, you may be rejecting real burial. Another way of looking at this would be to see how many samples are one-sigma and two-sigma above constant exposure, since presumably the ratios shouldn't be allowed to be that high and hence might be giving you a more realistic picture of statistical spread. Similar to the above, I think this piece is drawing conclusions that will be misleading to the reader. There is no magical cut-off here, and I feel strongly this should be treated more cautiously and with more transparency.

Response: We agree with the reviewer that this aspect was not sufficiently described. As the scientific debate on the matter indicates, burial is a difficult parameter to constrain—for similar reasons to those regarding inherited nuclides noted above. This also applies to our model. However, we do not subjectively pick any model based on the $^{26}\text{Al}/^{10}\text{Be}$ ratio. We circumvent the

issue entirely by modelling all samples irrespective of $^{26}\text{Al}/^{10}\text{Be}$ ratio. We then focus our discussion upon the resultant total erosion history, which is a robust outcome of our model that is not dependent on specifically identifying the existence or absence of burial. We also note that the MCMC approach taken here actually provides a quantitative sensitivity test of both burial and erosion. We agree with the reviewer that such tests are generally needed when interpreting CN data, and we therefore base our study upon them.

C2-6. Transferability: The ideas you propose here are interesting, and they provide clarity about your landscape of interest. But the paper does not explicitly address how transferable the approach and findings are to other landscapes (which may help with the “big picture”, see above). Is what you learned specific to Norway? Could it also apply to Greenland? What about cold-based ice landscapes in Antarctica? What factors might dictate why or why not your conclusions are transferable? You get there to some extent on Lines 185-188, but I would like to hear more.

Response: We expect similar patterns on other glaciated passive margins, although perhaps earlier in the glacial history, as continued warm-based glacial erosion tends to develop landscapes optimized for ice-discharge (e.g. large glacial troughs carrying most of the ice flux), thereby often leaving inter-trough areas with thinner, cold-based ice, with less power to erode the substratum. We have strengthened this aspect in the text (lines 169-182 and 204-216)

C2-7. Blanks: Your blanks are confusing since it sounds like you prepared samples in three separate labs and analyzed them on two different AMS systems. This needs to be made much more transparent for the reader. In the Methods, please detail exactly how many process blanks you have from which sets of conditions, and how you corrected. For example, were samples prepped at SUERC and analyzed at SUERC corrected using only the blanks from the same set of conditions? It's not clear as written. Further, the backgrounds you report seem to only be for the Aarhus AMS, not for SUERC, and you refer to them as “machine blanks”. This term usually implies a direct precip blank that has not been processed along with your samples. I assume you also ran process blanks and used the process blanks to correct the samples? This section needs to be clearer and the blank corrections may need some rethinking. I acknowledge the blank correction is quite small proportionally for these samples, but it still should be made clear to the reader.

Response: Thank you for giving us this chance to clarify our processing blank correction approach, we agree that this issue was not sufficiently documented. Each sample was blank-corrected with the processing blank that followed each batch of samples, i.e. processed under the same set of conditions. Processing blank data for each sample is reported in Supplementary table 2. This table also shows where each sample was processed and measured. We have added to the Methods section information on the processing blank levels, the blank:sample ratios and the magnitude of processing blank correction in percent. The machine blanks are, as you point out, directly precipitated blank material, that do not contain information on the laboratory process. We did not use these values to correct our sample ratios.

Minor comments:

C2-8. Line 21: Add how many measurements you made; it's an impressive number but is buried

about 80 lines into the paper.

Response: We agree and have added this number to the abstract.

C2-9. Line 41: I'm not sure "assumed" is the right word here since many of these studies are based in field observations and data. Maybe "inferred"?

Response: We have removed this sentence

C2-10. Line 43: Should be "suggests" instead of "suggest" since it's a singular study.

Response: Yes, we have changed this.

C2-11. Line 43: I don't think "significant" is the right word here. It's vague, and it also implies some sort of statistical meaning. I'd prefer to see an actual rate (or range of rates?).

Response: We have included the estimate of early plateau-erosion rates from the referenced study in order to make this more specific.

C2-12. Lines 45-49: I agree with you that your work is important for understanding sediment fluxes. But the sediment flux piece seems to come out of nowhere, since the abstract and the whole first paragraph are focused on landscape development rather than sediment generation. See the major comment about finding a coherent big picture.

Response: We have rephrased this paragraph so it shows a better link between erosion rates and sediment flux.

C2-13. Lines 61-63: Again, I agree with you that this is important for thinking about sediment fluxes. But make the link back to the big picture for the reader here. Why should we care about sediment flux? Why is it important?

Response: See response to comment C2-3.

C2-14. Line 73: "Recent" is vague here- clarify.

Response: We have chosen to be vague here on purpose as the time period that can be constrained depends on the erosion rates.

C2-15. Line 76: I don't think you need the hyphen after exposure.

Response: Ok, we have removed it.

C2-16. Line 90: You refer the reader to the supplement for existing age control on the termination of the last glaciation, but at least tell the reader here where that age control is coming from. Is it radiocarbon?

Response: The age control is not in our supplementary information, but in the two compilation studies that we cite in this sentence. The compilations are based on all available data that constrain the deglaciation chronology (cosmo, OSL, radiocarbon, varves etc.). We have changed the sentence to make this clearer.

C2-17. Line 115: Here and throughout, your inferred erosion rates need to have uncertainties. At present, you give the reader no sense for how well-constrained these are and what assumptions have gone into the calculations.

Response: We agree with the reviewer here and have added the interquartile range of the modelled distributions to give an indication of how well-constrained the median values we state are. However, in most other places where we refer to modelled values, we describe either a range or minimum values, and in those cases, we find it unnecessary to add uncertainties. Uncertainty associated with each modelled value is stated in Table 4 and also shown in Fig. 4.

C2-18. Line 118: Clarify how you define “overlapping”- indistinguishable at one-sigma?

Response: There is no clear definition here, because this is not a choice we make. The Monte Carlo algorithms search for models that fit the data, and we note that when the apparent exposure age gets close to the deglaciation age any model with a high erosion rate fits the data, and hence we cannot resolve the maximum rate. We have clarified this in the text.

C2-19. Line 124: Is it really fair to compare your erosion rates to all non-glacial settings? The Portenga paper looks at erosion rates all around the world, from a huge variety of climates. I’m not saying this comparison can’t be made, but it really needs to be clarified for the reader. I seem to remember that Portenga sub-divided based on climate classifications. How do your rates compare with other cold landscapes? And is the value you quote on line 125 just bedrock, or both bedrock and basins?

Response: We make the comparison to all non-glacial settings intentionally. The point is to stress that although the glacial erosion rates on plateaus are much smaller than in troughs, they are not generally small when compared to the rates of non-glacial surface processes acting on passive margins. The value we quote is for bedrock outcrops, which we have clarified in the text.

C2-20. Line 128: Should be “occurs”.

Response: OK – has been changed.

C2-21. Lines 137-140: See earlier comment about needing uncertainties on all of these rates.

Response: These values are minimum values referring to a number of samples, therefore we do not think that we need to include uncertainties here.

C2-22. Line 140: Clarify what “average” this is. For where? Is this averaged over the whole landscape, i.e. both plateaus and troughs?

Response: We are referring to erosion rates on the plateaus from a study by Steer et al. 2012. We have added ‘plateau’ to the sentence to specify this.

C2-23. Line 160-162: I’m struggling with how to interpret this sentence. The “elsewhere” is vague, and I’m having trouble with the 1 Myr time period. This is less than two half-lives for ^{26}Al , so I’m not clear on why you’re drawing a cut-off so recently. Do you mean that all the erosion on the plateaus is old?

Response: “Elsewhere” refers to other (previously) glaciated margins where plateau landforms have often been interpreted as old (sometimes tertiary) features because of old cosmogenic ages and low $^{26}\text{Al}/^{10}\text{Be}$ ratios. We have specified this. The 1 Myr time period was referring to the maximum “minimum limiting exposure and burial ages” from cosmogenic-nuclide studies from plateaus around the North Atlantic – ages in e.g. Antarctica are often much older. We have removed this part of the sentence.

C2-24. Line 169: "Recent" here is vague.

Response: We have changed this to late Cenozoic.

C2-25. Line 180: It's unclear what the 45-65% is. Is this the portion of plateau sediment contribution in relation to the whole?

Response: Yes, we have made that clearer.

C2-26. Lines 216-220: I like these ideas- they make an intriguing, broad hook for your paper. See my major comment about the big picture. As a reader, it was frustrating to wait until the last few sentences of the paper to see the larger significance, and seemed like an inconsistent message because much of the introduction is focused on sediment yields. The paper would flow more effectively if there were a more focused, coherent message throughout.

Response: As highlighted by the two other reviews, the discussion regarding long-term landscape evolution of the plateaus in Scandinavia (noted in the last paragraph of the manuscript) is controversial. We agree with reviewer #2 that this debate is highly interesting, however, given that we cannot gain insights of early or Pre-Pleistocene evolution from cosmogenic nuclides, we prefer not to over-stretch our observations, and so we do not wish to frame our paper around this debate. See also responses to C1-3 and C2-3.

C2-27. Line 286: When you refer to minimal shielding, do you mean topographic?

Response: Yes, we have changed the text to reflect this.

C2-28. Lines 306-307: I would like to see a few more sentences about your ^{27}Al quantification.

How did you assess the accuracy of your ICP work? Did you use an internal standard? How well do the expected and actual ^{27}Al contents of your blanks compare? Your ^{26}Al data are only ever as good as your ^{27}Al quantification, so these are important details to include for the reader.

Response: Thanks, this is an important point and we have accordingly added more details on our quality control of these measurements in the Methods section and added some more information to table 2.

C2-29. Figure 2: I assume the colored dots in the second panel are the ^{10}Be ages, color-coded the same way as in the first panel? Consider rewording the caption and potentially moving the legend so that it's clear what those dots are.

Response: Yes, it is the same legend, we have clarified that in the caption.

C2-30. Figure 4: For panel A, clarify in the caption that the ^{18}O threshold is something you've chosen and imposed upon the model. For panel B, what are the blue bar and orange circle on the right side? For panel C, you might specify which axes the different parameters are plotted against since it's a complex figure.

Response: A: The $\delta^{18}\text{O}$ threshold is a free parameter in the model and not something we choose explicitly, although we do stipulate a range of possible variation. The uncertainty is indicated by the red lines to each side. We have clarified this in the caption. B: The circle and bar represent the modelled median erosion rate and associated uncertainty (1st-3rd quartile of distribution) for this sample. We have clarified this in the caption. C: We have specified this in the caption.

Reviewer #3 (Remarks to the Author):

C3-1. This paper presents results from paired ^{10}Be and ^{26}Al cosmogenic isotope analyses for bedrock surfaces (60) and boulders (9) from glacially-eroded terrain that stands above the magnificent glacial trough of the Sognefjord and its tributaries. The main conclusions of the study are that (1) high-elevation plateaus can form as a result of glacial and periglacial processes within the Quaternary, and (2) need not be relict landforms inherited from a pre-Quaternary landscape and (3) do not represent dissected remnants of once conjoined, preglacial surfaces that have been left largely unmodified by ice sheets throughout the Quaternary.

The authors present important new data on the depths and rates of glacial erosion for a swath of terrain standing above the Sognefjord and rising W-E. This is a product of a bold sample campaign which yielded an unusually large cosmogenic isotope sample set. The paper is clearly written and the diagrams are generally very good. I suggest however that the new results do not adequately support the conclusions.

The concept of selective linear glacial erosion is now almost 50 years old¹. David Sugden recognised that in the Cairngorms Mountains, Scotland, ice sheets had covered the terrain and selectively carved deep troughs out of a pre-existing surface which, in immediately adjacent areas, it has left barely modified. On the Cairngorm plateau surfaces, landforms of glacial erosion such as roches moutonnées and rock basins are absent from large areas and non-glacial landforms, notably tors, are common, rising out slopes that also carry blockfields and, locally, saprolites. In contrast, in the Cairngorm glens glacial erosion has been highly effective, with deepening of precursor fluvial valleys by a maximum of 350 m (ref 2). Glacial erosion here has been selective and linear because it was at least an order of magnitude greater in depth along the valleys.

Sugden elegantly linked form to process. He reasoned that to excavate deep glacial valleys the ice moving within the valleys must be erosive whereas the preservation of fragile tors on plateaux required that here ice was non-erosive through multiple glacial cycles. He identified the pressure melting point of basal ice as a fundamental threshold for glacial processes. Ice above the PMP could slide and erode whereas ice below the PMP remained frozen to its bed and could not slide and erode, with movement confined to internal deformation and creep.

The Cairngorms became the type area for selective linear glacial erosion because of its striking morphological contrasts. Areas with similar geomorphology were soon identified from the passive margins around the North Atlantic³ and from the Canadian Arctic⁴. Landscapes of selective linear glacial erosion were recognised also as representing a distinctive zone of glacial erosion on passive margins, intermediate in position and erosion depths between the high elevation, intensely dissected rift shoulders of western Scotland and Norway, the west and east coasts of Greenland and the east coast of Labrador and the low elevation, low erosion zones of NE Scotland⁵, Finnish Lapland⁶, northern Greenland⁷ and Baffin Island⁸. Low erosion rates indicated by cosmogenic nuclides^{7,9} allow that in these landscapes of selective linear glacial erosion plateaux can be largely relict landforms inherited from a pre-Quaternary landscape and so may represent dissected remnants of once conjoined, preglacial surfaces that have been left largely unmodified by ice sheets throughout the Quaternary¹⁰.

This reasoning is broadly matched by arguments presented on Lines 30-39 and 145-155 in the manuscript. On Line 40 however there is a jump to a wider, rather different concept of selective linear glacial erosion. Here we start to deal with landscapes where glacial erosion is widespread – with glacial erosion forms widely developed outside of deep valleys. In the case of the Sognefjord, we are in a zone of high glacial erosion on the shoulder of the Norwegian passive margin, with extensive glacial scour outside of the glacial valleys. Glacial erosion is still very uneven in its impact in such landscapes^{11,12}, with at least one order of magnitude difference between erosion depths in the valley and those on ridges and hills¹³. Selective erosion occurs because glacier ice flows towards valleys and hollows and is thicker in those topographic lows and hence glacial erosion is concentrated in these lows. This selectivity drives the glacial buzzsaw to cut down preferentially in mountainous terrains¹⁴. Mathematical modelling shows how topography acting on glacial dynamics gives contrasts not only in basal ice temperatures but also in sliding velocities and meltwater fluxes within and outside valleys¹⁵.

The Sognefjord is the largest fjord system in Norway and penetrates about 200 km inland from the coast¹⁶. The mountains along Sognefjord rise eastward from about 500 m in the coastal region to reach elevations above 2000 m in Jotunheimen. The glacial valley now occupied by the fjord has been regarded as incised within an older valley form with an original relief of at least several tens of metres^{16,17}. Across SW Norway and within the Sognefjord basin, stepped erosion surfaces have been identified at elevations of 800-1200, 1300-1400, 1500-1800 and 2000 m a.s.l., each with internal relief of at least many tens of metres¹⁸. During the Pleistocene, ice caps and ice sheets built up repeatedly across the Sognefjord drainage basin, filling the Sognefjord and advancing onto the shelf. During the last glaciation, ice advance occurred at ~34 ka and the Sognefjord basin was probably ice covered for all of the period until ~11 ka, apart from a brief period of marine conditions at the outer coast in the Allerød. The behaviour of ice in Sognefjord was probably highly dynamic in space and time due to the extreme water depth in the fjord and the very high snow accumulation rates on the surrounding high ground¹⁹.

Response: We would like to thank the reviewer for the comments. We agree with most of these statements, as they correspond with the review of previous work outlined in our manuscript. As reviewer #3 is no doubt aware, however, previous interpretations of landscape evolution based solely on morphological observations are subject to considerable limitations, since surface morphology is the product of lithological factors together with surface processes. In our study, we choose to focus on carefully constrained erosion rates. Differing interpretations of shapes preserved in the landscape have not led to major consensus over the past century and while our measurements are also subject to interpretation, we suggest that provision of quantitative data are a major step forward.

In addition to the three conclusions summarized by the reviewer, we wish to clarify some additional significant points deriving from our study. Two key findings worth emphasising are i) the magnitude of glacial erosion on the high plateaus around Sognefjord is non-negligible, and ii) we use these erosion rates to estimate plateau contribution to long-term sediment flux from this region. We think that these results are important, irrespective of the ongoing long-term landscape evolution discussion. See also responses to C1-3 and C2-26. In response to this comment, as well as comments from the other reviewers, we have strived to clarify this message in the manuscript.

C3-2. The paper sets up two linked hypotheses to test using erosion rates derived from cosmogenic nuclides: (1) that glacial erosion is minimal beyond glacial troughs and that (2) plateaux represent little modified preglacial landsurfaces. Does the experimental design of the study fully address these hypotheses?

The study area is the high elevation basement terrain that lies within a swath, 5-20 km wide, along the axis of the Sognefjord and its tributaries. Is this a representative case study area? No if comparisons are to be drawn with the Cairngorm type area or with similar terrains with few glacial landforms on surrounding plateaux. No also if the preservation of little-modified preglacial surfaces is to be examined because the best preserved remnants of such landscapes in Fennoscandia have been mapped previously in zones of lower glacial erosion outside the Sognefjord basin^{20,21}. But yes if the focus is on valley-upland erosion contrasts on a deeply dissected rift shoulder. And yes because minimal erosion has been claimed for the terrain above the Sognefjord valley²². Due to the high relief and precipitation in SW Norway compared to the rest of Fennoscandia, it is likely that the volumes of ice discharged via Sognefjord were amongst the highest for outlet glaciers and ice streams along the Norwegian margin. Hence the rates of glacial erosion on the benches and hills alongside the glacial trough can be expected to be amongst the highest for terrain in comparable locations.

Response: Slow erosion rates on high-latitude plateaus with limited imprints of glacial erosion have previously been demonstrated in multiple studies as noted by the reviewer. In this study, we are looking at erosion rates in a fjord landscape where both plateaus and valleys have an imprint of glacial erosion, but where limited erosion of the plateaus has nonetheless been inferred. We expect that our modelled erosion rates (and the pattern with elevation) are representative for the wider region of fjord and plateau landscapes in SW Norway. Our objective is not to use the Sognefjorden area to constrain erosion rates of all other plateaus, but to use the plateaus around Sognefjorden to study if plateau erosion rates are necessarily low. This is clarified in lines 169-181 and 218-222.

C3-3. The samples are grouped in 10 localities alongside the Sognefjord and its tributaries. Grouping is helpful, as the authors point out, because it provides a window on the variability of cosmogenic nuclide inventories within each area. Little justification however is given in the text for choosing these sample areas. This is odd given the hypotheses to be tested. Where are the sample sites in relation to the supposed valley floor, plateaux and palaeosurfaces? Nesje and others have indicated where they think the former Sogne valley floor was.

Others have mapped palaeosurfaces and relief types across SW Norway, including the Sogne basin^{23,20}. To test the two hypotheses above it might be expected that sampling would have focussed on these specific locations regarded previously as preglacial topographic remnants. Moreover to test if remnants of precursor topography remain sample sites might include those where the lowest glacial erosion could be predicted. For example, apparently well-preserved plateau fragments mapped previously. Or sites remote from the main valley axis or in lee locations for ice flow. This seems to me to be a blind-spot in the sampling strategy of the project.

Response: Our strategy was aimed at collecting samples from high plateaus in a transect along Sognefjorden to document any potential gradients in erosion rates across elevation as well as distance from the coast. In practice, we were limited by the accessibility of sites and by a large

band of gabbroic (non-quartz-bearing) rocks that runs through the inner part of the Sognefjorden area from NE to SW.

In several places, we tried to target some of the areas previously mapped as ‘paleosurfaces’ or ‘Paleic surfaces’, but the local lithologies prevented us from recovering enough quartz from samples to provide ^{10}Be measurements. Although we did not generally target previously mapped ‘paleosurfaces’, our samples are in several cases collected from areas identified as part of the ‘Paleic surface’ (Nesje and Whillans, *Geomorphology* 9 (1994) p. 33-45) (see Fig. 1 this document). Furthermore, based on field observations together with satellite imagery of the area (Google Earth, norgebilder.no), it is not our impression that our selected sample-sites are more or less prone to glacial erosion or more or less glacially sculpted than the region in general. In other words, we see our sample sites as being representative of the fjord and plateau landscapes of southern Norway.

The reviewer notes that others have mapped palaeosurfaces and relief types across SW Norway, listing two studies in particular (Etzelmüller et al. 2007, Green et al., 2013) alongside the criticism that they might have served as reference points for our sampling strategy. We agree that reliable and detailed maps of the palaeosurfaces would have been extremely useful; however, no such maps exist on the scale required for our purpose—as we note below.

Etzelmüller et al. divides southern Norway into 10 highly generalised landform categories (their Fig. 7), denoting ‘high paleic mountains with glacial incisions, mostly moderate slopes’ on the highest parts of the Sognefjord basin and thereby covering most of our sample-sites (this category also covers the alpine topography in Jotunheimen). Another category: ‘Glacially scoured low mountains and valleys’ covers our sample-sites closer to the coast (Gulen and Lavik).

Green et al. present an even more generalised and schematic view of western Norway in their Fig. 4. It is not possible to use their oblique sketch map reliably for direct comparison with our sample-sites (i.e., it certainly cannot be transferred to a GIS). Furthermore, their ‘High-level-plains’ and ‘Inclined re-exposed surface’ fails to clarify what they consider to be a ‘paleosurface’. In any case, we have effectively sampled both “types”.

In summary, while we have indeed sampled the full range of morphologies described by others, neither of the references noted above provide a useful baseline map of the palaeosurfaces and we look forward to seeing one published in the future.

Fig. 1. Sample locations in this study in relation to the 'paleic surface' outlined by Nesje and Whillans, *Geomorphology* 9 (1994).

C3-4. Sample locations are on ridge and hill summits; not all locations can be described as belonging to plateaus, despite repeated, rather vague references to plateaus in the text. Local relief varies from tens to hundreds of metres within the sample areas at scales of a few km². Most locations have been scoured by ice - western locations tend to be glacially streamlined whereas eastern locations are more or less roughened. However morphological differences between areas and sites seen at different scales and elevations in air and ground photos are not assessed in relation to erosion depths in the paper – these differences may be important²⁴. Some sites retain blockfields, potentially a useful indicator of low intensity glacial erosion if rock weathering in frost-susceptible rock types is not entirely post-glacial, but I don't see where in the paper or supplementary information the blockfield samples are identified or the blockfields described?

Response: As noted above, even studies directly concerned with the palaeosurfaces of southern Norway do not provide detailed maps of these surfaces. We have clarified in the manuscript, which samples are from plateaus and which are not, but the definition of such plateaus is necessarily imprecise (see also response to C1-6). Furthermore, we have identified samples from our blockfield-site on figures and in the tables.

C3-5. What then of the results? Supplementary Figure 2 provides a clear summary of spatial distribution of the 10Be results. Cosmogenic isotope inheritance is present at all sites except perhaps for two western sites, Lavik and Balestrand. There is some doubt about whether or not slight nuclide inheritance is present because of the wide 9-12 ka range for deglaciation ages. Cosmogenic isotope inheritance requires that <2.5-3 m of rock was removed by the last ice sheet,

probably with <1 m at many sites. It would be good to see these calculations for the last glaciation to compare with the longer term modelled erosion rates.

Response: As noted in our response to Reviewer #2 (C2-4), we did not aim to draw a simple distinction between samples that contain nuclide inheritance and those that do not because our modelling approach makes this question irrelevant. The MCMC model addresses the question of inheritance in a systematic and objective manner by examining all possible scenarios within the model space and selecting those forward models that yield the best fit to our cosmogenic nuclide measurements. Targeting the erosion of the last ice sheet separately, as suggested by the reviewer, would on the other hand require assumptions regarding the glacial history of the sites as well as erosion rates during previous periods — assumptions that cannot be easily verified by observations. We have opted for a robust, albeit simple, analysis and we question whether the suggestion from this reviewer would yield any useful or additional information.

C3-6. For the westernmost site, Gulen, the presence of nuclide inheritance (shown by differences in bedrock and boulder ^{10}Be ages) is remarkable. This site is on a ridge transverse to ice flow and fully exposed to ice leaving the outer fjord. Yet <3 m of rock was removed from the ridge after ~20 ka of km-thick, warm-based ice flow. Indeed the whole data set is remarkable because it demonstrates widespread cosmogenic isotope inheritance across the Sogne basin, a major outlet for the last FIS.

Response: Bedrock and boulder ^{10}Be ages from Gulen overlap within 1 sigma, when using the external uncertainties from the CRONUS calculator (referenced in the paper) on individual samples. However, if we ignore uncertainties arising from scaling of production rates (which is a valid assumption for closely-spaced samples), the difference between apparent exposure ages of bedrock (weighted mean 13.6 ± 0.3 kyr) and boulder erratics (weighted mean 12.1 ± 0.4 kyr) becomes significant. Nonetheless, we favour an alternative explanation to that of the reviewer. We consider the apparent inheritance in bedrock at Gulen to be caused by a short-lived retreat and re-advance of the ice-margin towards the end of the last deglaciation—an event documented from sites just south of Sognefjorden (Mangerud et al., *Quaternary Science Reviews* 132 (2016) p.175-205; and Mangerud et al. 2016, *Boreas*, DOI: 10.1111/bor.12208). The simplest explanation is that the difference in cosmogenic nuclide inventories measured in bedrock and the boulder erratics is due to exposure during this period—and not an exceptional case of surface preservation as suggested by the reviewer.

The Markov-Chain Monte Carlo (MCMC) model approach developed by Knudsen et al.²⁵ is innovative and important – it allows extrapolation of cosmogenic isotope results back in time. MCMC is based on the assumption that the exposure/burial history can be divided into two distinct regimes: (i) glacial intervals with subglacial erosion and, due to shielding by the overlying ice sheet, no exposure, and (ii) interglacial intervals with subaerial erosion and full exposure, assuming no significant shielding by for example, snow, till, or vegetation. The rates of glacial and interglacial erosion may vary spatially, but for any particular bedrock sample the two erosion rates are assumed to be uniform throughout all glacials and interglacials, respectively. The MCMC model does not include abrupt individual erosion events, such as subglacial plucking, but integrates the effects of such events over time.

C3-7. In this study, the complication of snow cover is acknowledged in the text and addressed in the supplementary information. A likely error of 8% based on modern snowfall data (Line 356) is suggested, representing >1 m snow cover for 6 months of the year. This may be too conservative for high elevation sites and for periods of cooler climates than present. Greater snow shielding gives more significant over-estimation of erosion rates.

Response: Long-term snow distribution patterns for both this study area and for production rate calibration sites are mostly unknown and somehow accounting for it inevitably leads to unverifiable assumptions. Instead, we selected sites on topographic highpoints to minimize the effect, yet, we acknowledge that some of our sites might be buried beneath snow cover during winter. Based on this comment, we have included an additional snow-shielding estimate of 2 m of snow shielding for 8 months a year (i.e., ~20 % reduction in production rates). We consider this scenario to represent the maximum likely extent of shielding due to snow at our sample sites.

C3-8. Also MCMC assumes that the last glacial phase provides a close analogue for previous phases. This may be true but we know that FIS has switched between ice cap and ice sheet modes through the Pleistocene²⁶ and we suspect that ice in the Sognefjord drainage basin may behave very differently when the fjord is filled with ice or water.

Response: The assumptions involved in the MCMC approach is pointed out and discussed in lines 203-215 and are described in detail by Knudsen et al. (2015). Yes, the model uses a simple first approximation of complex erosion histories, but it still presents a major step forward from previous approaches — many of which fail to accommodate glacial erosion at all (e.g., Beel et al., 2016).

C3-9. A further complication is that glacially-polished rock surfaces probably are subject to increasing weathering and erosion rates after exposure.

Response: Rates of postglacial weathering are dependent upon a whole range of factors including lithology, aspect, soil and biotic interactions, temperature, and moisture availability. The reviewer's assertion that glacially-polished surfaces are subject to faster erosion (e.g., microgelivation) is contrary to published observations (e.g., André et al., 2002) and so we question that this assumption can be made.

C3-10. It should also be noted that the apparent ¹⁰Be ages all fall within the last 50 ka so that an extrapolation to time spans of 1 Ma suggested by expressing erosion rates in units of m/Myr is a long stretch. The modelled erosion estimates are all for summits and modelled as 2-93 m/Myr. The highest estimates are for samples with low TCN inheritance where the last ice sheet removed >2 m of rock. 73% of samples appear to indicate erosion rates of <30 m/1 Myr. Even at low elevation sites there are samples with rates <15 m/Myr. Erosion rates of <10 m/Myr are recorded from all sites at >800 m elevation. This latter point is crucial is because when viewed across extensive rock surfaces and at the Myr timescale it is the slowest erosion rates that matter. If nearby rock surfaces erode at, say, 10 m/Myr and 50 m/Myr then this would lead to increasing relief and leave the first surface upstanding. This topographic relief is not seen on sample summit areas. Also if erosion rates of 50 m/Myr were extensive then all summit sites with slow erosion rates of <10 m/Myr would be destroyed within 1 Myr. The differences in erosion rates within individual area likely relate mainly to what happened to different rock surfaces during the last glaciation – for example whether or not an episode of bedrock quarrying occurred.

Response: We find that large local variations occur between erosion-rate estimates for different samples in the same vicinity. This indicates that our modelled rates are sensitive to the timing of the last erosion event. We infer from this that it is necessary to take multiple samples from a site in order to gain an overall estimate approaching the ‘true’ average local erosion rate. To this end, we take the mean, rather than the minimum, as a better estimate of the site erosion. We acknowledge that the number of samples collected within this study is probably not sufficient to completely resolve the ‘true’ site erosion rates. However, our dataset is among the largest collections of new ^{26}Al and ^{10}Be data ever presented within a single study, and we have targeted sites that we expect to be less affected by glacial erosion (viz. small topographic highs rather than basins) in order to obtain a minimum estimate of surface lowering.

We acknowledge that, as pointed out by the reviewer, the erosion-rate integration timescale for many samples is far less than 1 Myr. However, we have tested these estimates against integration time (Fig. 2) and shown that rates generally stabilise after a few hundreds of thousand years (i.e. a few glacial cycles). Hence, it is appropriate to report these erosion rates in terms of m/Myr.

Figure 2. Median (black), quartiles (red) and 5 & 95% percentiles (magenta) as a function of integration period for a typical sample (SF33).

C3-11. Rates of erosion of <10 m/Myr are compatible with the modification of pre-existing valley floors and erosion surfaces with an original relief of many tens of metres. We should remember however that the erosion rates reported here come from summits in hard, relatively unfractured rock. Deepening of surrounding shallow, fracture-guided basins and valleys has involved greater depths and rates of glacial erosion. Hence glacial erosion in successive glaciation can act to lower, modify and destroy older topography. Glaciers tend to destroy rather than create low-relief rock surfaces through the innate tendency for ice flow to be faster, thicker and warmer along valleys¹⁴. Several mentions are made in the text of the formation of plateaux by glacial erosion but relief along the swath sampled in this paper is increasing – the basins and tributary valleys set into the terrain above the Sognefjord are becoming deeper.

Response: Glaciers can both enhance and reduce relief depending on the length-scale of the topography, as recently demonstrated by Egholm et al. (2017). On shorter topographic scales ice

can thus act to smoothen topography, while it increases relief on the basin-scale by carving out glacial troughs. Both of these effects should be considered when discussing the role of glacial erosion on landscape evolution and relief generation. See also response to C1-3.

C3-12. The authors are correct to point to significant glacial modification of rock surfaces in the uplands surrounding the Sognefjord. The glacial morphology developed across much of the Sognefjord basin, except its highest ground, makes this manifest. Similar modification has been mapped widely across SW Norway²⁰. In the lowland basement terrain of southern Sweden, where extensive palaeosurfaces are also mapped²⁷, Pleistocene glacial erosion of basement is estimated generally as 10-20 m below summits, reaching >50 m in valleys²⁸. We may be seeing the start of a convergence of FIS glacial erosion long-term average rates in hard bedrock at a magnitude of ~10 m/Myr. The results presented in this paper thus may represent an important contribution to the rejection of the extreme and rather dated end-member views of no Pleistocene modification of palaeosurfaces versus complete glacial eradication of palaeosurfaces presented at the start of this paper. The way is clear for re-adoption of an older, more moderate view^{29,30} that widespread topographic inheritance is possible if significant but spatially variable glacial and periglacial erosion is accepted.

Response: We agree that topographic inheritance is possible, e.g. in the fluvial network valley structure and we have strived to make this clearer (see also response to C1-3). However, the term 'palaeosurface' has a loaded connotation in the Scandinavian context, where Mesozoic erosion surface formation at sea level and later uplift is often inferred.

C3-13. The question of sediment volumes offshore remains. The volume of rock evacuated from the Sognefjord was first generated by Nesje and others from a smoothed summit surface using summit heights and then subtracting the present relief from it. This surface subsumes small valleys and rock basins and so this erosion is included in estimates. To this should be added the summit erosion depths estimated by the authors. Perhaps also an allowance for former saprolite layers³¹. That will still leave a large deficit between source and sink – pointing to deep erosion of the coast and the near-shore zone³².

Response: In the manuscript, we point to the possibility that the average erosion rates were higher and more uniform during early glacial periods, before the landscape was transformed to optimize transport of ice (e.g. large U-shaped troughs). However, we agree that extensive erosion of the coast and near-shore zone is another possibility for converging sediment flux estimates during this period. More research is needed to test these hypotheses. This would require erosion estimates on plateaus and in the coastal zone over timescales that exceed what can be obtained with currently established methods. Cosmogenic nuclides are too shallow, whereas thermochronological methods have largely been inconclusive with regards to the late Cenozoic evolution of topography in Scandinavia because the long-term erosion is too slow.

Adrian Hall

References

- 1 Sugden, D. E. The selectivity of glacial erosion in the Cairngorm Mountains, Scotland. Transactions of the Institute of British Geographers 45, 79-92, doi:10.2307/621394 (1968).
- 2 Hall, A. M. & Gillespie, M. Fracture control on valley persistence: the Cairngorm Granite pluton,

- Scotland. *International Journal of Earth Sciences*, doi:10.1007/s00531-016-1423-z (2016).
- 3 Sugden, D. E. Landscapes of glacial erosion in Greenland and their relationship to ice, topographic and bedrock conditions. *Institute of British Geographers Special Publication 7*, 177-195 (1974).
- 4 Sugden, D. E. Glacial erosion by the Laurentide ice sheet. *Journal of Glaciology* 20, 367-391 (1978).
- 5 Hall, A. M. & Sugden, D. E. Limited modification of mid-latitude landscapes by ice sheets: the case of north-east Scotland. *Earth Surface Processes and Landforms* 12, 531-542, doi:10.1002/esp.3290120510 (1987).
- 6 Hall, A. M., Sarala, P. & Ebert, K. Late Cenozoic deep weathering patterns on the Fennoscandian shield in northern Finland: a window on ice sheet bed conditions at the onset of Northern Hemisphere glaciation. *Geomorphology* 246, 472-488 (2015).
- 7 Corbett, L. B., Bierman, P. R., Graly, J. A., Neumann, T. A. & Rood, D. H. Constraining landscape history and glacial erosivity using paired cosmogenic nuclides in Upernavik, northwest Greenland. *Geological Society of America Bulletin* 125, 1539-1553 (2013).
- 8 Ebert, K. GIS-analyses of ice-sheet erosional impacts on the exposed shield of Baffin Island, eastern Canadian Arctic. *Canadian Journal of Earth Science*, doi:10.1139/cjes-2015-0063 (2015).
- 9 Corbett, L. B., Bierman, P. R. & Davis, P. T. Glacial history and landscape evolution of southern Cumberland Peninsula, Baffin Island, Canada, constrained by cosmogenic ^{10}Be and ^{26}Al . *Geological Society of America Bulletin* 128, 1173-1192, doi:10.1130/b31402.1 (2016).
- 10 Schermer, E. R., Redfield, T. F., Indrevær, K. & Bergh, S. G. Geomorphology and topography of relict surfaces: the influence of inherited crustal structure in the northern Scandinavian Mountains. *Journal of the Geological Society*, doi:10.1144/jgs2016-034 (2016).
- 11 Anderson, R. S., Molnar, P. & Kessler, M. A. Features of glacial valley profiles simply explained. *Journal of Geophysical Research* 111, 1-14 (2006).
- 12 Kessler, M. A., Anderson, R. S. & Briner, J. P. Fjord insertion into continental margins driven by topographic steering of ice. *Nature Geoscience* 1, 365-369 (2008).
- 13 Strunk, A. et al. One million years of glaciation and denudation history in west Greenland. *Nature Communications* 8, 14199, doi:10.1038/ncomms14199 <http://www.nature.com/articles/ncomms14199#supplementary-information> (2017).
- 14 Hall, A. M. & Kleman, J. Glacial and periglacial buzzsaws: fitting mechanisms to metaphors. *Quaternary Research* 81, 189-192, doi:10.1016/j.yqres.2013.10.007 (2014).
- 15 Hall, A. M. & Glasser, N. F. Reconstructing former glacial basal thermal regimes in a landscape of selective linear erosion: Glen Avon, Cairngorm Mountains, Scotland. *Boreas* 32, 191-207, doi:10.1080/03009480310001100 (2003).
- 16 Nesje, A. & Sulebak, J. R. Quantification of late Cenozoic erosion and denudation in the Sognefjord drainage basin, western Norway. *Norsk Geografisk Tidsskrift* 48, 85 - 92 (1994).
- 17 Aarseth, I., Nesje, A. & Fredin, O. West Norwegian Fjords. 45 (Norsk Geologisk Forening, 2014).
- 18 Lidmar-Bergström, K., Bonow, J. M. & Japsen, P. Stratigraphic Landscape Analysis and geomorphological paradigms: Scandinavia as an example of Phanerozoic uplift and subsidence. *Global and Planetary Change* 100, 153-171 (2012).
- 19 Mangerud, J., Gyllencreutz, R., Lohne, Ö. & Svendsen, J. I. in *Quaternary Glaciations -Extent and Chronology: Part IV. a closer look.* (eds J. Ehlers & P. Gibbard) 279-298 (Elsevier, 2011).
- 20 Etzelmüller, B., Romstad, B. & Fjellanger, J. Automatic regional classification of topography in Norway. *Norsk Geologisk Tidsskrift* 87, 167-180 (2007).

- 21 Fjellanger, J. & Etzelmüller, B. Stepped palaeosurfaces in southern Norway - interpretation of DEM-derived topographic profiles. *Norsk Geografisk Tidsskrift* 57, 102-110 (2003).
- 22 Nesje, A., Dahl, S. O., Valen, V. & Ovstedal, J. Quaternary erosion in the Sognefjord drainage basin, western Norway. *Geomorphology* 5, 511-520, doi:10.1016/0169-555X(92)90022-G (1992).
- 23 Green, P. F., Lidmar-Bergström, K., Japsen, P., Bonow, J. & Chalmers, J. A. Stratigraphic landscape analysis, thermochronology and the episodic development of elevated, passive continental margins. *Geological Survey of Denmark and Greenland Bulletin* 30, 1-150 (2013).
- 24 Krabbendam, M. & Bradwell, T. Quaternary evolution of glaciated gneiss terrains: pre-glacial weathering vs. glacial erosion. *Quaternary Science Reviews* 95, 20-42 (2014).
- 25 Knudsen, M. F. et al. A multi-nuclide approach to constrain landscape evolution and past erosion rates in previously glaciated terrains. *Quaternary Geochronology* 30, 100-113, doi:10.1016/j.quageo.2015.08.004 (2015).
- 26 Kleman, J., Stroeven, A. P. & Lundqvist, J. Patterns of Quaternary ice sheet erosion and deposition in Fennoscandia and a theoretical framework for explanation. *Geomorphology* 97, 73-90, doi:10.1016/j.geomorph.2007.02.049 (2008).
- 27 Lidmar-Bergström, K., Olvmo, M. & Bonow, J. M. The South Swedish Dome: a key structure for identification of peneplains and conclusions on Phanerozoic tectonics of an ancient shield. *GFF*, 1-16, doi:10.1080/11035897.2017.1364293 (2017).
- 28 Lidmar-Bergström, K. A long-term perspective on glacial erosion. *Earth Surface Processes and Landforms* 22, 297-306, doi:10.1130/G20343.1 (1997).
- 29 Linton, D. L. The forms of glacial erosion. *Transactions of the Institute of British Geographers* 33, 1-28, doi:10.2307/620998 (1963).
- 30 Rudberg, S. Gross morphology of Fennoscandia - six complementary ways of explanation. *Geografiska Annaler* 70A, 135-167, doi:10.2307/521068 (1988).
- 31 Glasser, N. F. & Hall, A. M. Calculating Quaternary erosion rates in North East Scotland. *Geomorphology* 20, 29-48, doi:10.1016/S0169-555X(97)00005-6 (1997).
- 32 Hall, A. M., Ebert, K., Kleman, J., Nesje, A. & Ottesen, D. Selective glacial erosion on the Norwegian passive margin. *Geology* 41, 1203-1206, doi:10.1130/G34806.1 (2013).

REVIEWERS' COMMENTS:

Reviewer #1 (Remarks to the Author):

I really enjoyed reading this revised paper! It brings a wealth of new cosmogenic data to quantify the role of glaciers in shaping passive margins in maritime locations. It is clear and very well presented. I think it will stand as an important benchmark for the way Cenozoic glaciations in the northern hemisphere modify passive margins.

My concern before was that the original paper seemed to test a hypothesis that was not strictly relevant to what they had found. They have completely changed this in response to all three referees and now the paper directly approaches the problem of how much modification has taken place in a maritime setting. The result is new and fascinating insight, for example an excellent discussion of why their results differ from those based on sediment accumulation. This latter discussion raises lots of questions and new hypotheses and seems just the way science should proceed.

Assuming the authors have dealt with the technical issues raised by the other referees, I fully support publication of the paper in its present form.

Two very minor points.

Line 195. For smoother English omit what has and been and replace with that ?

Line 220. Add after plateaus : in a maritime setting or in western Scandinavia. Just emphasize once more in the conclusion that they are studying a maritime margin? There is exceptionally good evidence further east in more continental Sweden, where the ice was cold-based, of less erosion.

I am happy for my name to be made known!

David Sugden

Reviewer #3 (Remarks to the Author):

The authors have made a careful revision of the manuscript and thereby produced, in this reader's view, a more balanced, nuanced and focussed study. The paper presents important and detailed new evidence from cosmogenic nuclides for the glacial erosion of high plateaux in W Norway. The paper represents a major contribution to the literature on the glacial erosion of passive margins.

The arguments in the paper are presented clearly and the diagrams are also clear. I see no typos. I congratulate the authors on a job well done. Adrian Hall

Second review. Reviewer comments are in black with responses from the authors in blue.

Reviewer #1 (Remarks to the Author):

I really enjoyed reading this revised paper! It brings a wealth of new cosmogenic data to quantify the role of glaciers in shaping passive margins in maritime locations. It is clear and very well presented. I think it will stand as an important benchmark for the way Cenozoic glaciations in the northern hemisphere modify passive margins.

My concern before was that the original paper seemed to test a hypothesis that was not strictly relevant to what they had found. They have completely changed this in response to all three referees and now the paper directly approaches the problem of how much modification has taken place in a maritime setting. The result is new and fascinating insight, for example an excellent discussion of why their results differ from those based on sediment accumulation. This latter discussion raises lots of questions and new hypotheses and seems just the way science should proceed.

Assuming the authors have dealt with the technical issues raised by the other referees, I fully support publication of the paper in its present form.

Two very minor points.

Line 195. For smoother English omit what has and been and replace with that ?

Line 220. Add after plateaus : in a maritime setting or in western Scandinavia. Just emphasize once more in the conclusion that they are studying a maritime margin? There is exceptionally good evidence further east in more continental Sweden, where the ice was cold-based, of less erosion.

I am happy for my name to be made known!
David Sugden

We would like to thank David Sugden for the very helpful and encouraging comments, we have incorporated the suggested changes.

Reviewer #3 (Remarks to the Author):

The authors have made a careful revision of the manuscript and thereby produced, in this reader's view, a more balanced, nuanced and focussed study. The paper presents important and detailed new evidence from cosmogenic nuclides for the glacial erosion of high plateaux in W Norway. The paper represents a major contribution to the literature on the glacial erosion of passive margins.

The arguments in the paper are presented clearly and the diagrams are also clear. I see no typos. I congratulate the authors on a job well done. Adrian Hall

We thank Adrian Hall for the highly constructive and helpful suggestions that have helped to clarify the objectives of the study.